# Stabilization of V1 interneuron-motor neuron connectivity ameliorates motor phenotype in a mouse model of ALS

Santiago Mora[1,2], Anna Stuckert[1,2,5], Rasmus von Huth Friis[1,5], Kimberly Pietersz [3], Gith Noes-Holt [1], Roser Montañana-Rosell [1], Haoyu Wang[2], Andreas Toft Sørensen [1], Raghavendra Selvan [4], Joost Verhaagen[3] & Ilary Allodi [1,2] ✉

Loss of connectivity between spinal V1 inhibitory interneurons and motor neurons is found early in disease in the SOD1$^{G93A}$ mice. Such changes in premotor inputs can contribute to homeostatic imbalance of motor neurons. Here, we show that the Extended Synaptotagmin 1 (*Esyt1*) presynaptic organizer is downregulated in V1 interneurons. V1 restricted overexpression of Esyt1 rescues inhibitory synapses, increases motor neuron survival, and ameliorates motor phenotypes. Two gene therapy approaches overexpressing *ESYT1* were investigated; one for local intraspinal delivery, and the other for systemic administration using an AAV-PHP.eB vector delivered intravenously. Improvement of motor functions is observed in both approaches, however systemic administration appears to significantly reduce onset of motor impairment in the SOD1$^{G93A}$ mice in absence of side effects. Altogether, we show that stabilization of V1 synapses by *ESYT1* overexpression has the potential to improve motor functions in ALS, demonstrating that interneurons can be a target to attenuate ALS symptoms.

Somatic motor neurons are the ultimate output of the brain since they control movements by directly connecting to muscles. During locomotion their synchronized or reciprocal activation is regulated by a complex network of inhibitory and excitatory spinal interneurons known to regulate flexor-extensor muscle coordination, left-right alternation, rhythm and speed of locomotion[1]. Hence, functional connectivity between motor neurons and their premotor circuits is a prerequisite for maintenance of inhibitory-excitatory balance and execution of movements. In the fatal disease Amyotrophic Lateral Sclerosis (ALS), somatic motor neurons degenerate, and subjects progressively lose the ability to perform movements. In our previous work[2], we showed that the spinal V1 inhibitory interneurons, positive for Engrailed-1 (*En1*) transcription factor, lose their synapses onto the vulnerable fast-twitch fatigable motor neurons

early in disease in the SOD1$^{G93A}$ ALS mouse model[3]. This preferential loss of inhibitory inputs onto fast-twitch fatigable motor neurons might contribute to their unbalanced excitability[4], leading to excitotoxicity and ultimately to their vulnerability to disease. Moreover, V1 inhibitory interneurons are known to control the speed of locomotion in vertebrates[5], and the loss of connectivity observed in the SOD1$^{G93A}$ mice led to an onset of locomotor phenotype. Such phenotype is characterized by a reduction of speed and acceleration, a decrease in stride length and step frequency, and a hyperflexion of the hindlimbs[2]. These symptoms were observed at a timepoint preceding motor neuron death and muscle denervation, and could be directly associated to loss of V1 inputs[2,5,6]. Changes in V1 excitability and electrophysiological properties were observed as early as postnatal day 6 in the SOD1$^{G93A}$ mice[7]. Further evidence of alterations

[1]Department of Neuroscience, University of Copenhagen, Copenhagen, Denmark. [2]School of Psychology and Neuroscience, University of St Andrews, St Andrews, UK. [3]The Netherlands Institute for Neuroscience, Amsterdam, The Netherlands. [4]Department of Computer Science, University of Copenhagen, Copenhagen, Denmark. [5]These authors contributed equally: Anna Stuckert, Rasmus von Huth Friis. ✉e-mail: ia51@st-andrews.ac.uk

in synaptic connectivity has been reported in ALS[8]. Loss of inhibitory synapses has been described in a Fused in sarcoma (Fus) mouse showing Amyotrophic Lateral Sclerosis and Frontotemporal Dementia (FTD)-like phenotypes[9,10]. Thus, inhibitory synaptopathy is not restricted to the SOD1[G93A] mouse model. Interestingly, synaptic proteomics performed on postmortem tissue of C9ORF72 patients identified ~500 proteins with altered expression levels also within inhibitory synapses[11]. Moreover, two recent studies showed that misprocessing of *UNC13A* mRNA strongly associates with ALS-FTD pathology caused by TDP43 downregulation[12,13]. The *UNC13A* gene plays a pivotal role in neurosecretion and is a fundamental component of neuron-to-neuron communication[14,15], suggesting general synaptic dysregulations in the disease. Thus, the potential development of strategies directed to promote neurosecretion and synapse stabilization might be beneficial in the attempt to overcome such synaptopathy.

The Extended synaptotagmin 1 (Esyt1) is a presynaptic protein previously shown to promote synaptic growth and stabilization[16], and is preferentially expressed in neurons resistant to ALS[17]. Here, we show that *Esyt1* is downregulated early in disease in neurons found within the ventral horn of the spinal cord and in V1 interneurons. In the present study, we investigated if by stabilizing connectivity between spinal V1 inhibitory interneurons and motor neurons we could modify disease progression in the SOD1[G93A] mice. To this aim, we overexpressed the human *ESYT1* specifically in V1 interneurons using a cre-lox gene therapy approach. *ESYT1* overexpression was found to (1) increase the number of bona fide inhibitory synapses on motor neurons, (2) rescue motor neuron loss and (3) ameliorate motor phenotypes in the treated SOD1[G93A] mice. In this report we show improvement of motor functions in ALS induced by a spinal interneuron-restricted treatment. Moreover, our results point toward

Esyt1 and inhibitory synapses being putative targets for future investigations on disease modifiers.

## Results

### The extended synaptotagmin 1 (*Esyt1*) transcript is down-regulated early in disease

We investigated the expression of the *Esyt1* transcript by RNAscope in the lumbar segments of the spinal cord (L1-L4) of control and SOD1[G93A] mice at three different timepoints, postnatal day (P) 45, 63 and 84 (Fig. 1). These timepoints were chosen based on our previous findings[2] which showed significant loss of glycinergic inputs onto motor neurons at P45 and loss of En1 transcript at P63 and P84. *Esyt1* expression was found specifically in spinal neurons including motor neurons (Fig. 1a). Firstly, quantifications were performed in the ventral horn of the spinal cord in control and SOD1[G93A] tissue (Fig. 1b, c). 4–12 hemi sections per mouse, 5-6 mice per condition and timepoint randomly chosen areas of the ventral spinal cord were quantified per mouse. Downregulation was observed already at P45, as well as P63 and 84; the same timepoints at which we found decreased levels of the *En1* transcript[2] (Fig. 1d) (Two-way ANOVA and Tukey's post hoc, P45 $P = 0.0051$, P63 $P = 0.013$, P84 $P = 0.0046$, $N = 5$-6 per timepoint, per genotype). Secondly, *Esyt1* expression was investigated specifically in *En1* positive neurons with RNAscope. Here, the number of double positive neurons was counted in control and SOD1[G93A] tissue (Fig. 1e, f). Quantifications demonstrated downregulation of *Esyt1* transcript in V1 interneurons starting from P63 (Fig. 1g) (Two-way ANOVA and Tukey's post hoc, P63 $P = 0.0004$, P84 $P = 0.0025$, $N = 3$-6 per timepoint, per genotype). Quantifications are shown per hemi section. Thus, the presynaptic transcript *Esyt1* is downregulated early during disease progression in the SOD1[G93A] mice in neurons within the ventral horn of the spinal cord and in V1 interneurons.

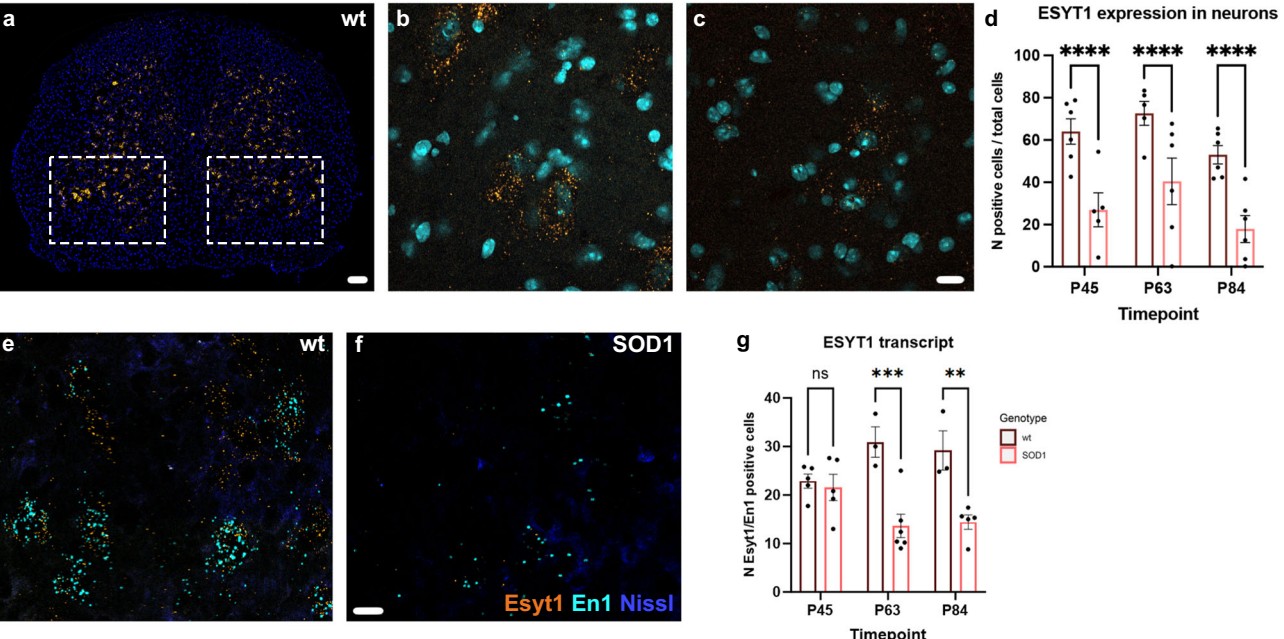

**Fig. 1 | *Esyt1* downregulation in SOD1[G93A] mice. a** Expression of *Esyt1* transcript in the lumbar spinal cord of a WT mouse visualized by RNAscope. Dashed boxes indicate quantified areas. **b** 40X magnification microphotograph showing *Esyt1* expression in the ventral horn of the spinal cord in a WT mouse. **c** *Esyt1* is downregulated in neurons during early ALS progression in the ventral horn of the spinal cord (*Esyt1* in orange, DAPI in light blue). **d** Quantifications performed in WT and SOD1[G93A] mice at P45, P63 and P84 show a downregulation of *Esyt1* transcript already at P45 (Two-way ANOVA and Tukey's post hoc, P45 $P = 0.0051$, P63 $P = 0.013$, P84 $P = 0.0046$, WT P45 $N = 6$, SOD1[G93A] P45 $N = 5$, WT P63 $N = 5$, SOD1[G93A]

P63 $N = 6$, WT P84 $N = 6$, SOD1[G93A] P84 $N = 6$). When looking specifically at *En1*+ interneurons, (**e**) expression of *Esyt1* transcript is found in *En1* positive neurons in WT mice, (**f**) but it is downregulated in the SOD1[G93A] mice; microphotographs show *Esyt1* expression at P84. **g** Quantifications performed in WT and SOD1[G93A] mice at P45, 63 and 84 show a downregulation of *Esyt1* transcript in *En1*+ cells starting at P63 (Two-way ANOVA and Tukey's post hoc, P63 $P = 0.0004$, P84 $P = 0.0025$, WT P45 $N = 5$, SOD1[G93A] P45 $N = 5$, WT P63 $N = 3$, SOD1[G93A] P63 $N = 6$, WT P84 $N = 3$, SOD1[G93A] P84 $N = 5$). Scale bar in **a** = 100 μm, in **b–c** = 10 μm and in **e–f** = 20 μm. All graphs show mean values ± SEM. Source data are provided as a Source Data file.

## Gene therapy driven *ESYT1* overexpression increases Esyt1 protein in interneurons

To investigate if rescue of *ESYT1* expression could restore connectivity between inhibitory interneurons and motor neurons, we overexpressed *ESYT1* specifically in V1 interneurons by viral delivery. To achieve V1 restricted overexpression, an adeno associated (AAV) serotype 8 virus was generated to overexpress *ESYT1* upon *cre*-dependent recombination. The AAV8-hSYN-DIO-hESYT1-W3SL virus was injected intraspinally in SOD1[G93A] mice crossed with En1[cre] mice[18] (Fig. 2a, b). Phenotype and genotype of the double transgenic SOD1[G93A];En1[cre] mice, including copy number of the mutated gene, was evaluated and did not differ from mice expressing SOD1[G93A] alone (Supplementary Fig. 1a–c), hence all SOD1[G93A] mice included in the study showed the same disease progression. All four genotypes resulting from the crossing – WT, En1[cre], SOD1[G93A] and SOD1[G93A];En1[cre] – received bilateral injections in each of the L1-L3 lumbar segments (six in total) of 100 nl

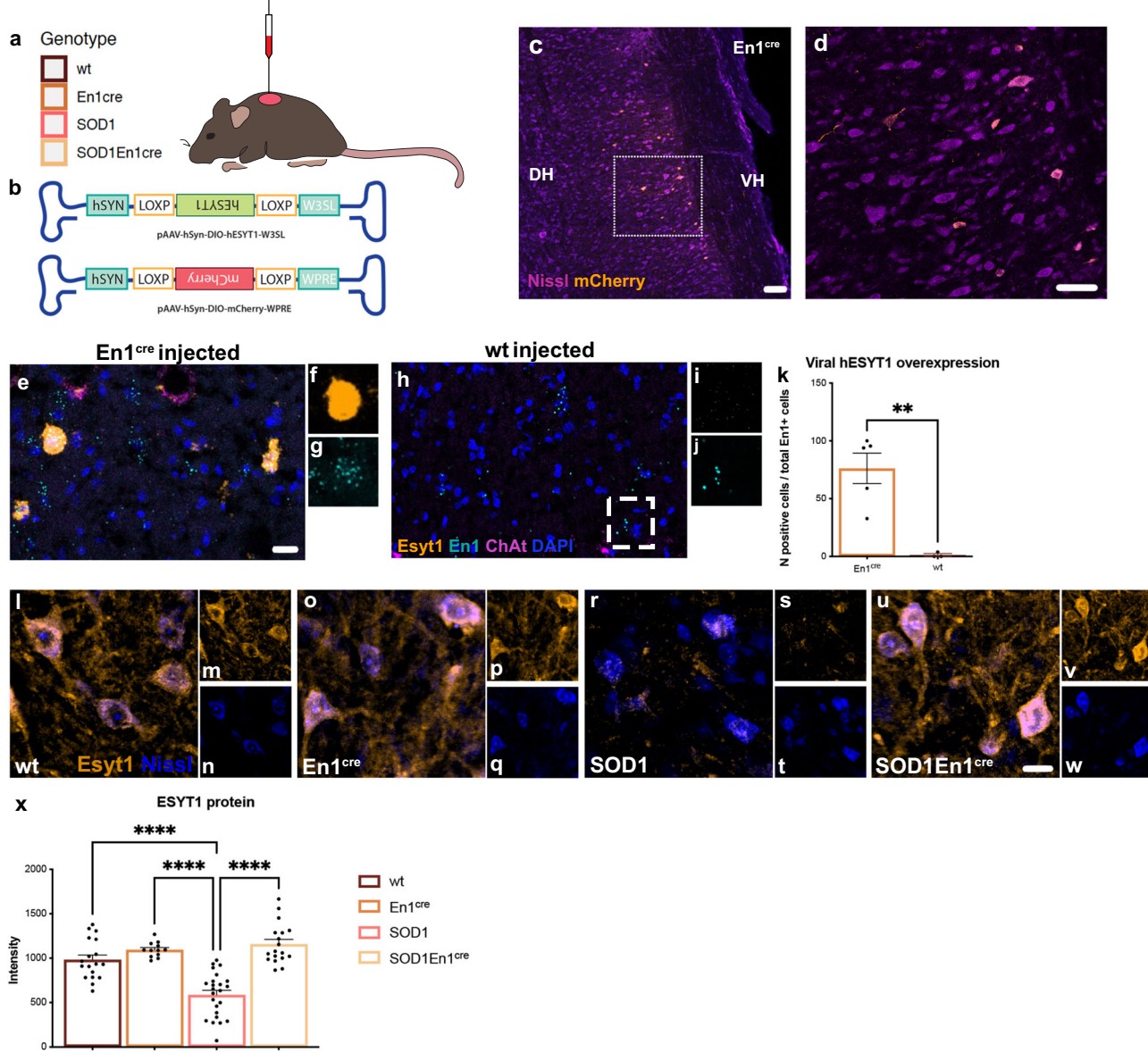

**Fig. 2 | Cartoon depicting the methodological approach for the AAV intraspinal delivery experiments. a** Intraspinal injections were performed in WT, En1[cre], SOD1[G93A] and SOD1[G93A];En1[cre] mice, all genotypes were injected in L1-L3 spinal segments. **b** Schematic of cre-dependent constructs designed to overexpress *ESYT1* and *mCherry*. **c–d** Tile reconstruction of a longitudinal section of the spinal cord of an En1[cre] mouse upon overexpression of the AAV8-hSYN-DIO-mCherry virus validates successful cre-dependent expression in the ventral/medial areas of the cord. AAV8-hSYN-DIO-hESYT1-W3SL driven overexpression was analyzed by RNAscope utilizing a probe recognizing the inverted viral construct upon cre-recombination, **e** Expression of viral-h*ESYT1* in En1[cre] mice. **f** Close-up microphotograph showing a neuron overexpressing *ESYT1*. **g** The same neuron is also positive for *En1* transcript. **h** *ESYT1* was not detected in WT injected mice. **i, j** *En1* positive neuron negative for *ESYT1*. **k** Quantification of *ESYT1* positive neurons in the lumbar spinal cord of En1[cre] and WT mice, 76% of *En1+* neurons are positive for *ESYT1* (*t*-test, two-tailed, *P* = 0.0052, En1[cre] *N* = 5, WT *N* = 3). **l–w** Esyt1 protein expression in interneurons found within the ventral horn of the spinal cord in the four experimental conditions. **l–n** Esyt1 expression in WT mice; (**o–q**) Esyt1 overexpression in En1[cre] healthy mice; (**r–t**) Esyt1 is downregulated in the SOD1[G93A] mice and (**u–w**) expression is rescued after AAV8-hSyn-DIO-hEsyt1-W3SL delivery in SOD1[G93A];En1[cre] mice. **x** Quantifications of Esyt1 intensities in interneurons found within the ventral horn of the spinal cord at P112 in all four experimental conditions show increased Esyt1 protein expression in the SOD1[G93A];En1[cre] mice (One way ANOVA and Fisher's LSD post hoc *P* = 0.003; WT = 3, En1[cre] *N* = 2, SOD1[G93A] *N* = 4, SOD1[G93A];En1[cre] *N* = 3; 10–20 interneurons analyzed per condition). Scale bar in **c** = 100 μm, in **d** = 50 μm, in **e**, **h** = 20 μm and in **l**, **o**, **r**, **u** = 20 μm. All graphs show mean values ± SEM. Source data are provided as a Source Data file.

each at postnatal day 30. Virus was used at a final titer of $4 \times 10^{12}$ vg/mL. L1-L3 segments were targeted since they are the first affected in the SOD1[G93A] mice[19], and responsible for the onset of locomotor phenotype[2]. An AAV8-hSYN-DIO-mCherry-WPRE virus was also generated and injected as described above in the lumbar segment of the spinal cord of En1[cre] and SOD1[G93A];En1[cre] mice to validate V1-restricted transduction and overall viral spread (Fig. 2b–d). mCherry positive neurons were found ~1.2 mm rostrally and caudally from injection site (Fig. 2c). Due to the large size of the *ESYT1* insert (~3.3k bp), a fluorescent tag could not be added to the AAV8-hSYN-DIO-hESYT1-W3SL and *ESYT1* overexpression was analyzed utilizing an RNAscope probe recognizing the inverted vector sequence upon *cre*-recombination (Fig. 2e). As expected, overexpression was specific to neurons within the ventral/medial areas of the spinal cord and restricted to *cre* mice (Fig. 2e–j). Fluorescence was not detected in the WT injected mice (Fig. 2h–j). Manual quantification of h*ESYT1* revealed overexpression in 76 % of *En1*+ neurons in the analyzed lumbar segments (Fig. 2k) (*t*-test, *P* = 0.0052, En1[cre] *N* = 5, WT *N* = 3). Moreover, Esyt1 protein expression was investigated in interneurons within lamina VII, VIII and IX of the spinal cord of WT, En1[cre], SOD1[G93A] and SOD1[G93A];En1[cre] mice at P112 after AAV8-hSYN-DIO-hESYT1-W3SL administration (Fig. 2l–w). Interneurons were identified by their ventral and intermediate location within the spinal cord and their smaller soma size (average perimeter = ~50 μm, average area = ~150 μm²)[20], and pixel intensities were

quantified using confocal microscopy. An antibody recognizing Esyt1 was used, hence both endogenous and overexpressed ESYT1 proteins are visualized in this assay. Quantifications showed increased intensity for Esyt1 in SOD1[G93A];En1[cre] compared to untreated SOD1[G93A] (Fig. 2x) (One-way ANOVA and Fisher's LSD post hoc *P* = 0.003; *N* = 2–4 mice per condition; 20 interneurons analyzed per condition). Hence, these results demonstrate that our viral approach is specific for V1 interneurons and leads to increased expression of Esyt1 protein.

## Rescue of inhibitory synapses and increased motor neuron survival upon *ESYT1* overexpression

Upon AAV8-hSYN-DIO-hESYT1-W3SL administration in all four genotypes, changes in inhibitory synaptic density on motor neurons were investigated at P112 (Fig. 3a–e). This timepoint was chosen since significant motor neuron loss can be observed at P112 in the SOD1[G93A] mice[21]. We showed a decrease in inhibitory synaptic inputs at P45, P63 and P84 timepoints[2], however P112 was not previously investigated. Here, inhibitory synaptic inputs were visualized utilizing a Vgat antibody, while motor neurons were identified by ventral localization, size, and Chat staining (Fig. 3a–d, f–i). Vgat synaptic density on motor neurons was reconstructed, masks were obtained for every single motor neuron showing clear nucleus and corrected by their soma perimeter (Fig. 3a–d). Injected SOD1[G93A];En1[cre] mice exhibited significant increase in synaptic density when compared to

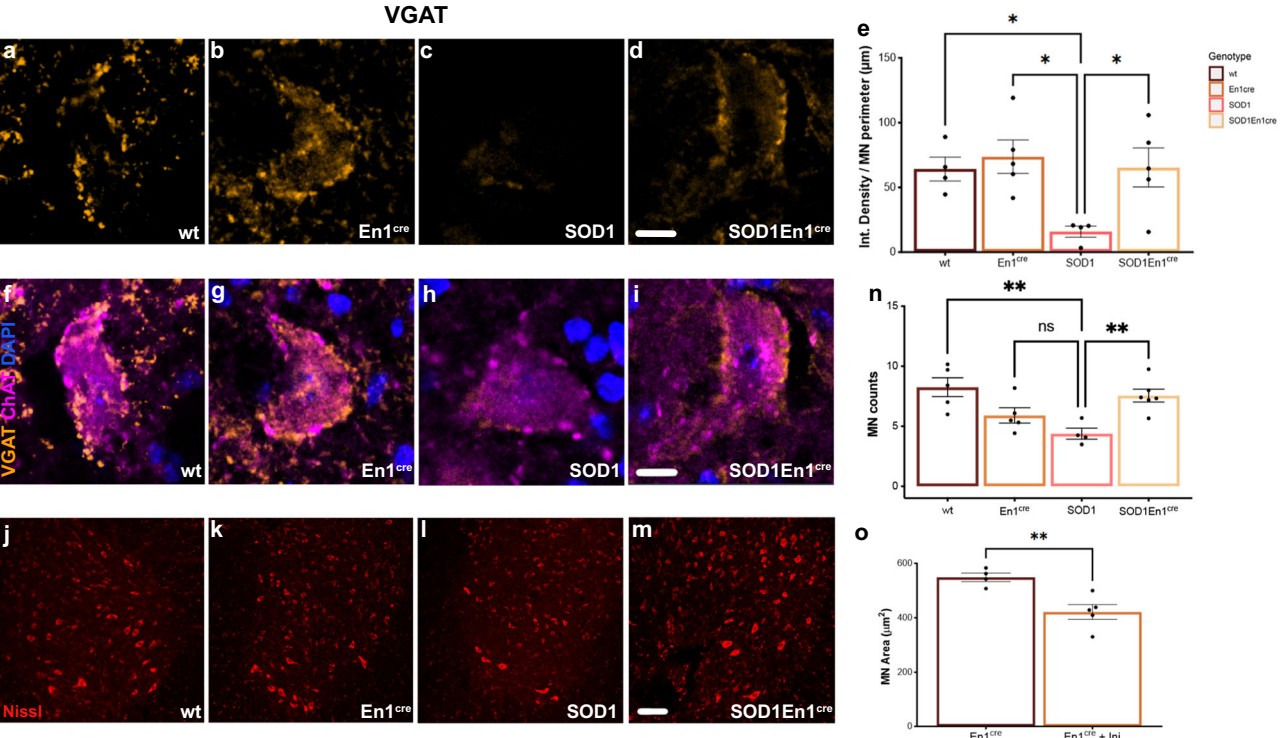

**Fig. 3 | *ESYT1* overexpression increases motor neuron survival and synaptic density on spared motor neurons.** Quantifications of inhibitory synapses on spared motor neurons at P112 upon AAV8-hSYN-DIO-hESYT1 injections. Synaptic densities are normalized by motor neuron perimeter. Examples of Vgat positive synaptic densities in the different conditions are shown in (**a–d**). **a, b** Show synaptic density in WT and En1[cre] mice respectively. **c** and (**d**) show differences in synaptic densities between SOD1[G93A] and SOD1[G93A];En1[cre] mice. **e** Synaptic density in SOD1[G93A];En1[cre] is significantly increased compared to SOD1[G93A] and comparable to the synaptic densities of WT and En1[cre] mice (One-way ANOVA and Dunnett's post hoc WT *P* = 0.045, En1[cre] *P* = 0.0119, SOD1[G93A];En1[cre] *P* = 0.0306, WT = 4, SOD1[G93A] *N* = 5, En1[cre] *N* = 4, SOD1[G93A];En1[cre] *N* = 5). Merge microphotographs in (**f–i**) show examples of quantified Chat positive motor neurons in the different genotypes. Vgat in orange, Chat in magenta, DAPI in blue. Motor neuron quantifications

performed at P112 in (**j**) WT, (**k**) En1[cre], (**l**) SOD1[G93A] and (**m**) SOD1[G93A];En1[cre] mice. Fluoro-Nissl in red. **n** SOD1[G93A];En1[cre] mice show increased motor neuron survival upon *ESYT1* overexpression when compared to the SOD1[G93A] mice (One-way ANOVA and Dunnett's post hoc SOD1[G93A]; En1[cre] *P* = 0.0080, WT *P* = 0.0023, *N* = 6). A minimum of 170 motor neurons per condition was quantified. En1[cre] mice overexpressing hEsyt1 show a trend in lower number of large motor neurons in the lumbar spinal cord (One-way ANOVA and Dunnett's post hoc, En1[cre] *P* = 0.2732, *N* = 5). **o** Comparison of motor neuron size in En1[cre] non-injected and En1[cre] injected mice with AAV8-hSYN-DIO-hESYT1 shows shrinkage of motor neurons upon *ESYT1* overexpression (*t*-test, two-tailed, *P* = 0.0073, En1[cre] non-injected *N* = 4, En1[cre] injected *N* = 5). Scale bar in **a–d** = 10 μm, **f–i** = 10 μm and in **j–m** = 50 μm. All graphs show mean values ± SEM. Source data are provided as a Source Data file.

injected SOD1$^{G93A}$, similar to injected WT and En1$^{cre}$ conditions (Fig. 3e). Motor neurons were also quantified at the same timepoint in all four AAV8-hSYN-DIO-hESYT1-W3SL injected genotypes (Fig. 3j–m). Here, only larger neurons over 28 µm in diameter within the ventral horn of the spinal cord were quantified. SOD1$^{G93A}$; En1$^{cre}$ mice showed an increased number of spared motor neurons compared to SOD1$^{G93A}$ mice after AAV8-hESYT1 overexpression (Fig. 3n) (One-way ANOVA and Dunnett's post hoc WT $P = 0.045$, En1$^{cre}$ $P = 0.0119$, SOD1$^{G93A}$;En1$^{cre}$ $P = 0.0306$, $N = 4$-6). 6–12 spinal cord hemi sections were quantified per mouse. A trend in reduction of larger motor neurons in En1$^{cre}$ control mice upon *ESYT1* overexpression was observed (Fig. 3n). Here, *ESYT1* is overexpressed in healthy V1 interneurons, hence overloading the cells with this protein which not only has a role in synaptic stabilization, but is also pivotal in the maintenance of membrane tethering and cell homeostasis. Analysis of motor neuron areas of control En1$^{cre}$ mice with and without *ESYT1* overexpression demonstrated a shrinkage of motor neurons at P112 (Fig. 3o). The molecular mechanisms involved in these maladaptive changes upon *ESYT1* overexpression in healthy neurons will require future investigations. Altogether these results demonstrate rescue of synaptic connectivity in the treated SOD1$^{G93A}$; En1$^{cre}$ mice as well as increased number of spared motor neurons.

## Esyt1 overexpression at presynaptic level in V1 interneurons increases the number of bona fide inhibitory synapses on motor neurons

To validate Esyt1 overexpression at synaptic level, simultaneous intraspinal injections of two viruses was performed (Fig. 4a). Here, the AAV8-hSYN-DIO-hESYT1-W3SL was mixed with the AAV-phSYN1(S)-FLEX-tdTomato-T2A-SypEGFP-WPRE, which, upon transfection of neurons, expresses eGFP under the synaptophysin promoter[22] (Fig. 4a), thus allowing for visualization of synapses (Fig. 4b–k). En1$^{cre}$ and SOD1$^{G93A}$;En1$^{cre}$ mice were injected ($N = 3$ per condition) and proximity ligation assay (PLA) was performed on transduced tissue 3 weeks after injections. The PLA allows for visualization of proteins located at 30 nm distance from each other upon double antibody binding. Here, anti-Gfp and anti-Esyt1 antibodies were used (Fig. 4a). Signal obtained from combined expression of Gfp and Esyt1 on motor neurons, identified by their localization in lamina IX and their size and shape, is shown in red at synaptic level in En1$^{cre}$ (Fig. 4b, f–h) and SOD1$^{G93A}$;En1$^{cre}$ (Fig. 4i–k) mice. The negative control shows staining in the absence of primary antibodies (Fig. 4c–e). Hence, these data confirms that Esyt1 is present at synaptic level upon AAV8-hSYN-DIO-hESYT1-W3SL overexpression. However, we next aimed to corroborate if the rescued

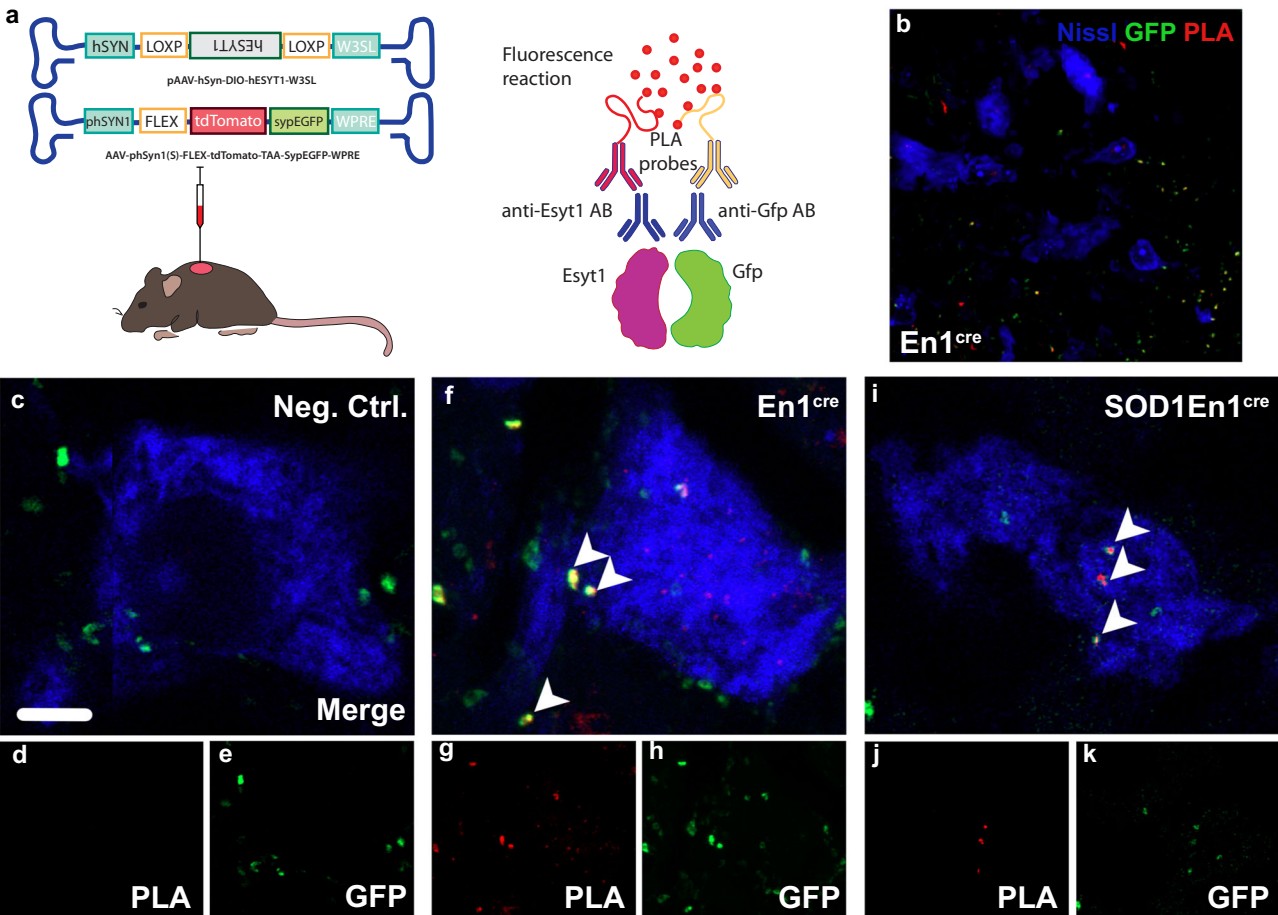

**Fig. 4 | Co-localization of Esyt1 at synaptic level after AAV8-hSyn-DIO-hEsyt1 and AAV-phSyn1(S)-FLEX-tdTomato-T2A-SypEGFP-WPRE administration in En1$^{cre}$ and SOD1$^{G93A}$;En1$^{cre}$ mice. a** Cartoon depicting experimental approach with dual virus injection and Proximity ligation assay (PLA). AAV8-hSyn-DIO-hEsyt1 and AAV-phSyn1(S)-FLEX-tdTomato-T2A-SypEGFP-WPRE were injected at 1:1 concentration. PLA recognizing Esyt1 and Gfp proteins was used to validate Esyt1 expression at synaptic level. **b** Lower magnification microphotograph shows Gfp targeting also other cells in the intermediate area of the spinal cord (GFP in green, PLA in red, Nissl in blue). **c–e** Negative control experiment without primary antibody incubation does not show PLA signal on motor neurons (**d**). **f–h** Three weeks after viral injection, and overexpression, Esyt1 is located synaptically on motor neurons in En1$^{cre}$ and (**i–k**) SOD1$^{G93A}$;En1$^{cre}$ mice. PLA reveals interaction of Esyt1 and Gfp antibodies at synaptic level after combined viral delivery of (in red, **g**, **j**), that co-localizes with Gfp signal alone (in green, **h**, **k**). Findings were replicated in two SOD1$^{G93A}$;En1$^{cre}$ and two En1$^{cre}$ mice. Images (**c–k**) are acquired at 63x magnification. Arrows indicate co-localization. Scale bar in **c** = 5 µm. Source data are provided as a Source Data file.

synapses were bona fide inhibitory synapses, presenting both the presynaptic and postsynaptic elements. Thus, we performed immunohistochemistry for Vgat and Gephyrin, which are proteins respectively found presynaptically and postsynaptically[23] (Fig. 5). Here, Vgat and Gephyrin expression on motor neurons was investigated in WT (Fig. 5d), SOD1$^{G93A}$ (Fig. 5i), En1$^{cre}$ (Fig. 5n) and SOD1$^{G93A}$;En1$^{cre}$ (Fig. 5s) at P112 upon Esyt1 overexpression. Scatter plots show the level of colocalization of Vgat and Gephyrin in the different conditions, highlighting the downregulation in SOD1$^{G93A}$ compared to the other genotypes (Fig. 5e–t). Quantifications show

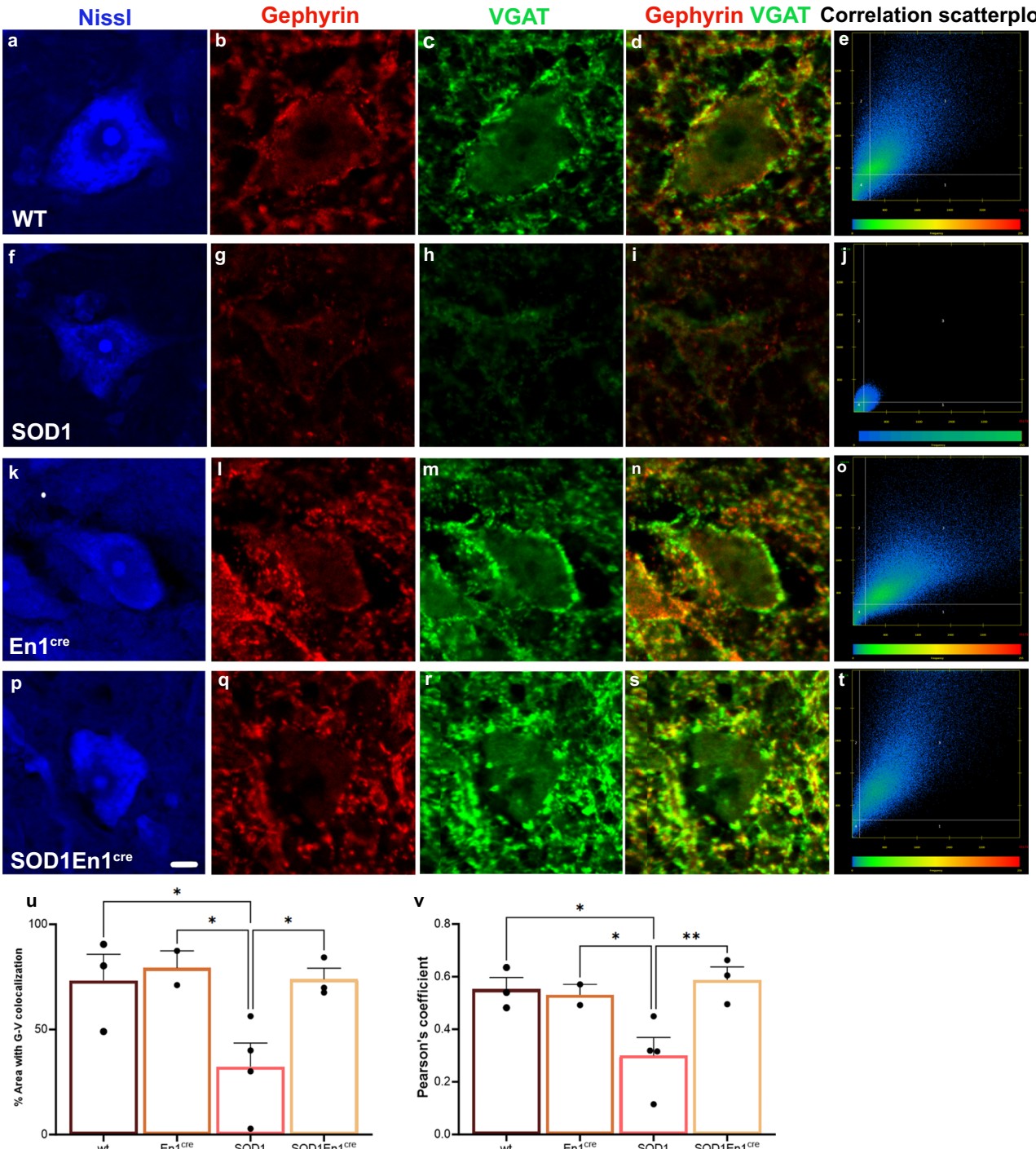

**Fig. 5 | *ESYT1* overexpression rescues bona fide inhibitory synapses on motor neurons. a–e** Inhibitory presynaptic marker Vgat and inhibitory postsynaptic marker Gephyrin co-localize in motor neurons of WT mice. **e** Co-localization of Vgat and Gephyrin is shown in the correlation scatterplot. **f–i** Both markers are downregulated in SOD1$^{G93A}$ mice at P112, (**j**) correlation scatterplot shows dramatic decrease of Vgat and Gephyrin co-localization in the SOD1$^{G93A}$ mice. **k–o** Co-localization of Vgat and Gephyrin in motor neurons of En1$^{cre}$ mice upon *ESYT1* overexpression. **p–t** Rescue of Vgat and Gephyrin expression in SOD1$^{G93A}$;En1$^{cre}$ mice overexpressing *ESYT1*. **t** Correlation scatterplot shows increased co-localization of Vgat and Gephyrin in the SOD1$^{G93A}$;En1$^{cre}$ condition. **u** Quantifications of the area where Vgat and Gephyrin are found co-localized in the different conditions (One-way ANOVA and Fisher's LSD post hoc Vgat: $P = 0.003$; Gephyrin: $P = 0.003$, WT = 3, En1$^{cre}$ $N = 2$, SOD1$^{G93A}$ $N = 4$, SOD1$^{G93A}$;En1$^{cre}$ $N = 3$). **v** Both pre- and postsynaptic inhibitory markers colocalize as indicated by Pearson's correlation; SOD1$^{G93A}$;En1$^{cre}$ mice show higher colocalization than SOD1$^{G93A}$ mice and similar to WT and En1$^{cre}$ controls (One-way ANOVA and Fisher's LSD post hoc: $P = 0.006$, WT = 3, En1$^{cre}$ $N = 2$, SOD1$^{G93A}$ $N = 4$, SOD1$^{G93A}$;En1$^{cre}$ $N = 3$). All graphs show mean values ± SEM. Source data are provided as a Source Data file.

the percentage of the area where Vgat and Gephyrin colocalize (One-way ANOVA and Fisher's LSD post hoc Vgat: WT $P = 0.009$, En1$^{cre}$ $P = 0.07$, SOD1;En1$^{cre}$ $P = 0.003$; Gephyrin: WT $P = 0.01$, En1$^{cre}$ $P = 0.008$, SOD1;En1$^{cre}$ $P = 0.003$; $N = 2$–4 mice per condition; 20 motor neurons analyzed per condition) (Fig. 5u) and the Pearson's correlation coefficient per condition (One-way ANOVA and Fisher's LSD post hoc: WT $P = 0.01$, En1$^{cre}$ $P = 0.03$, SOD1En1$^{cre}$ $P = 0.006$; $N = 2$–4 mice per condition; 20 motor neurons analyzed per condition) (Fig. 5v). Altogether these data demonstrate that *ESYT1* over-expression increases the number of bona fide inhibitory synapses in the SOD1$^{G93A}$;En1$^{cre}$ mice, compared to the untreated SOD1$^{G93A}$.

## *ESYT1* overexpression ameliorates motor phenotypes in the SOD1$^{G93A}$ mice

Next, we analyzed motor phenotypes after *ESYT1* overexpression in all four genotypes by placing the mice on a treadmill at a speed of 20 cm/

s, equivalent to a fast walk[24]. Videos were recorded from ventral and lateral views. Our previously published data showed that ~40% of SOD1$^{G93A}$ mice cannot cope with such speed by P63. Hence, we investigated if the increased synaptic connectivity upon *ESYT1* over-expression could ameliorate SOD1$^{G93A}$ motor phenotype. Following a brief training period, mice were assessed once a week from postnatal day P49 until P112. Videos were analyzed using the DeepLabCut marker-less pose estimation tool[25] as previously described[2]. SOD1$^{G93A}$ mice showed a significant reduction in locomotor performance and, consistently with our previous data, 37.5% showed an onset of loco-motor phenotype by P63 (Supplementary Fig. 2a). However, upon *ESYT1* overexpression only 14.3% of the SOD1$^{G93A}$; En1$^{cre}$ mice had an onset of locomotor phenotype by P63, and 57 % of them could perform the task by P112 (SOD1$^{G93A}$ median = P70, SOD1$^{G93A}$;En1$^{cre}$ median = P105). Here, average speed, step frequency, stride length and peak acceleration were analyzed longitudinally (Fig. 6a–d). SOD1$^{G93A}$;En1$^{cre}$

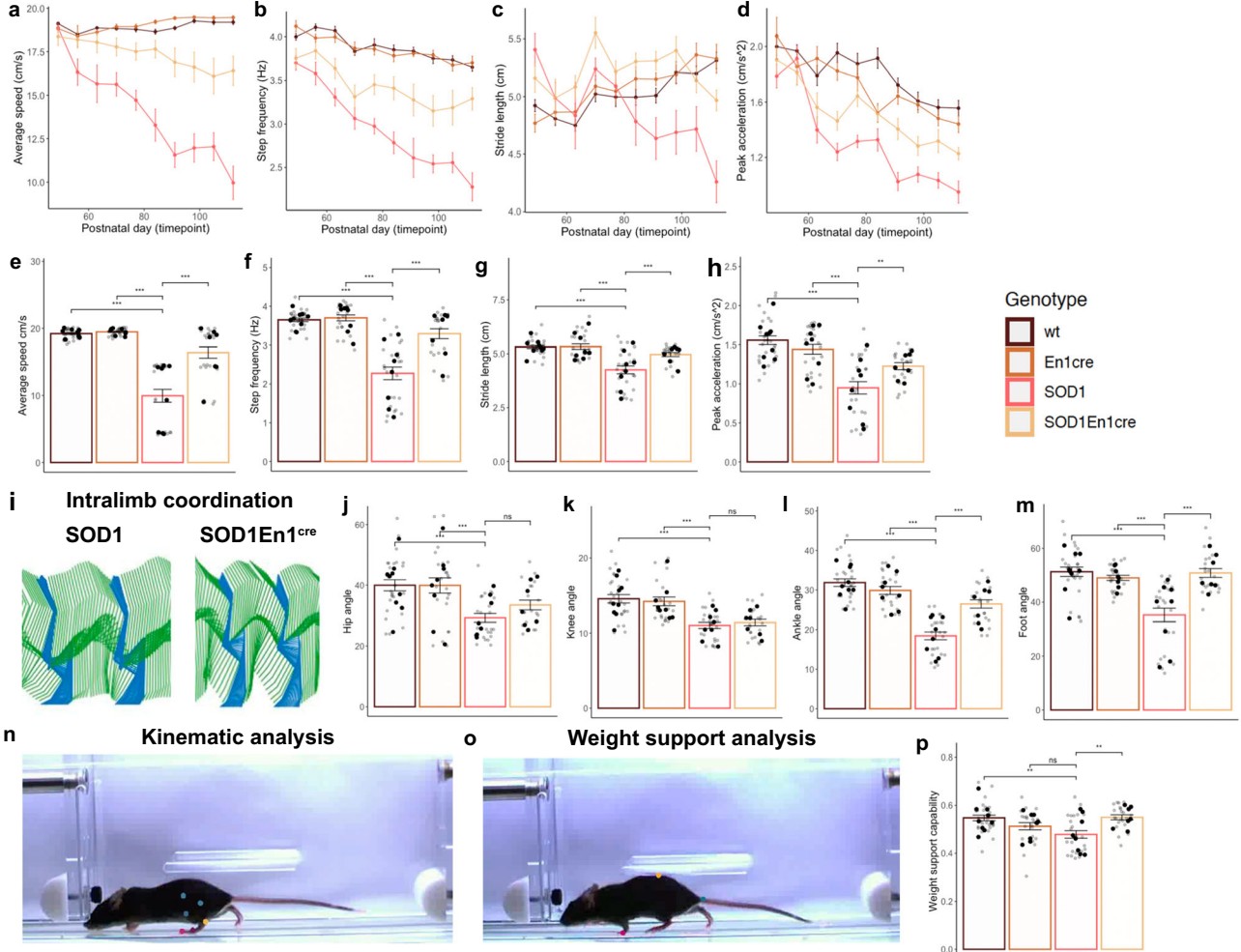

**Fig. 6 | Ameliorated locomotor phenotype upon *ESYT1* overexpression.**
**a** Average speed between day 49 and 112 shows amelioration in SOD1$^{G93A}$;En1$^{cre}$ mice from P56, (**b**) step frequency from P77, (**c**) stride length from P84, and (**d**) peak acceleration from P77. P49: WT $N = 12$, En1$^{cre}$ $N = 6$, SOD1$^{G93A}$ $N = 8$, SOD1$^{G93A}$;En1$^{cre}$ $N = 7$, P56: WT $N = 9$, En1$^{cre}$ $N = 4$, SOD1$^{G93A}$ $N = 7$, SOD1$^{G93A}$;En1$^{cre}$ $N = 6$, P63: WT $N = 11$, En1$^{cre}$ $N = 9$, SOD1$^{G93A}$ $N = 8$, SOD1$^{G93A}$;En1$^{cre}$ $N = 7$; P70-P105: WT $N = 10$, En1$^{cre}$ $N = 8$, SOD1$^{G93A}$ $N = 8$, SOD1$^{G93A}$;En1$^{cre}$ $N = 7$, P112: WT $N = 9$, En1$^{cre}$ $N = 8$, SOD1$^{G93A}$ $N = 8$, SOD1$^{G93A}$;En1$^{cre}$ $N = 7$. At P112 (**e**) speed is higher in SOD1$^{G93A}$;En1$^{cre}$ compared to SOD1$^{G93A}$ (One-way ANOVA and Dunnett's post hoc, SOD1$^{G93A}$;En1$^{cre}$ vs SOD1$^{G93A}$ $P = 3$e-10), as well as (**f**) step frequency (One-way ANOVA and Dunnett's post hoc, $P = 1.1$e-08), (**g**) stride length (One-way ANOVA and Dunnett's post hoc, $P = 5$e-04), and (**h**) peak acceleration (One-way ANOVA and Dunnett's post hoc, $P = 0.0087$) (WT $N = 9$; SOD1$^{G93A}$ $N = 8$, En1$^{cre}$ $N = 8$, SOD1$^{G93A}$;En1$^{cre}$ $N = 7$). (**i**) Stick figures depict

intralimb coordination. Two full cycles are visualized for SOD1$^{G93A}$ and SOD1$^{G93A}$;En1$^{cre}$ mice upon injections. Stance phase blue, swing phase green. Changes in joint angles were analyzed for (**j**) hip, (**k**) knee, (**l**) ankle and (**m**) foot angles. Significant changes were found for the ankle (One-way ANOVA and Dunnett's post hoc, $P = 8.5$e-07), and foot (One-way ANOVA and Dunnett's post hoc, $P = 2$e-07) angles upon *ESYT1* overexpression (WT = 9, SOD1$^{G93A}$ $N = 8$, En1$^{cre}$ $N = 8$, SOD1$^{G93A}$;En1$^{cre}$ $N = 7$). **n**, **o** Markers for kinematic and weight support analysis.
**p** Weight support is successfully recovered in SOD1$^{G93A}$;En1$^{cre}$ compared to SOD1$^{G93A}$ (One-way ANOVA and Dunnett's post hoc, $P = 0.0015$). All quantifications were performed in triplicates, graphs show mean values ± SEM, average values in (**e**–**h**, **j**–**m**, and **p**) are shown in black, technical triplicates in gray. Source data are provided as a Source Data file.

mice showed an amelioration in average speed from P56 (Two-way ANOVA and Dunnett's post hoc, $P = 0.0453$) (Fig. 6a), step frequency from P77 (Two-way ANOVA and Dunnett's post hoc, $P = 0.003$) (Fig. 6b), stride length from P84 (Two-way ANOVA and Dunnett's post hoc, $P = 0.0078$) (Fig. 6c) and peak acceleration from P77 (Two-way ANOVA and Dunnett's post hoc, $P = 0.0081$; $N = 4$-12 mice per condition; all quantifications were performed in triplicates) (Fig. 6d). When assessing the final timepoint (P112) SOD1$^{G93A}$;En1$^{cre}$ mice generally outperformed SOD1$^{G93A}$ mice, and half of them showed preserved average speed (One-way ANOVA and Dunnett's post hoc, SOD1$^{G93A}$;En1$^{cre}$ vs SOD1$^{G93A}$ $P = 3e$-10) (Fig. 6e). Amelioration was also observed for step frequency (One-way ANOVA and Dunnett's post hoc, $P = 1.1e$-08) (Fig. 6f), stride length (One-way ANOVA and Dunnett's post hoc, $P = 5e$-04) (Fig. 6g), and peak acceleration (One-way ANOVA and Dunnett's post hoc, $P = 0.0087$) (Fig. 6h) (WT $N = 9$; SOD1$^{G93A}$ $N = 8$, En1$^{cre}$ $N = 8$, SOD1$^{G93A}$;En1$^{cre}$ $N = 7$, all quantifications were performed in triplicates). Intralimb kinematics was performed by analyzing changes in degree amplitudes of four angle joints: hip, knee, ankle, and foot (Fig. 6n, Supplementary Fig. 3 and Supplementary Movie 1 and 2). 15 individual steps per trial were analyzed and the mean amplitude for each mouse was calculated. Hindlimb hyperflexion, a key element of the phenotype resulting from the loss of V1 synaptic inputs (Supplementary Fig. 3), was improved in the SOD1$^{G93A}$;En1$^{cre}$ mice upon h*ESYT1* overexpression for the foot (One-way ANOVA and Dunnett's post hoc, $P = 2e$-07) (Fig. 6m) and the ankle angles (One-way ANOVA and Dunnett's post hoc, $P = 8.5e$-07) (Fig. 6l), but not for the hip and the knee (Fig. 6j–k). The hip angle that was found unchanged at early stages of disease[2] is significantly reduced at P112 (Fig. 6j). Currently, we do not know the direct cause of these changes in the hip angle, however, V1 restricted-*ESYT1* overexpression does not seem to restore it (Fig. 6j). Moreover, comparisons of lateral view recordings between SOD1$^{G93A}$ and SOD1$^{G93A}$;En1$^{cre}$ mice revealed that the latter could better support their body weight as they maintained their bodies at a greater distance from the belt of the treadmill (One-way ANOVA and Dunnett's post hoc, $P = 0.0015$) (Fig. 6o, p, Supplementary Movie 1−kinematics of SOD1$^{G93A}$ and Supplementary Movie 2 – kinematics of SOD1$^{G93A}$;En1$^{cre}$). This suggests that the mice not only show ameliorated motor functions at the treadmill, but that force is maintained in the hindlimbs to counteract gravity.

## ESYT1 systemic administration obtained with an AAV-PHP.eB viral vector

Since intraspinal injections were only affecting lumbar segments of the spinal cord, survival studies were not performed in the AAV8-hSYN-DIO-hESYT1-W3SL injected mice because we considered it unlikely that such restricted delivery could impact the overall survival. Thus, to assess if *ESYT1* overexpression could prolong the life span of SOD1$^{G93A}$ mice, an AAV-PHP.eB overexpressing h*ESYT1* was generated (Fig. 7b). AAV-PHP.eB viruses can pass the blood brain barrier and transduce cells in the central nervous system upon intravenous delivery[26]. Here, three AAV-PHP.eB viruses were generated and delivered by retro-orbital injections (Fig. 7a): a cre-dependent AAV-PHP.eB-hSYN-DIO-hESYT1 delivered to SOD1$^{G93A}$;En1$^{cre}$ mice; a cre-dependent AAV-PHP.eB-hSYN-DIO-mCherry administered to En1$^{cre}$ mice to analyze the spread and specificity of the injections; and an AAV-PHP.eB-hSYN-eGFP delivered to SOD1$^{G93A}$ as control construct (Fig. 7b). Virus was injected at a concentration of $3.03 \times 10^{11}$ vg, and 100 μl were delivered per mouse at P30. WT mice were injected with saline. En1$^{cre}$ mice were euthanized at P70 to assess viral spread of the mCherry virus (Fig. 7c–h). Spinal cord and brain were sectioned and mCherry positive neurons were quantified in the lumbar spinal cord (Fig. 7c, d), midbrain (Fig. 7e, f) and somatosensory cortex (Fig. 7g, h) to investigate transduction efficiency and potential off targets. Quantifications show preferential targeting of the lumbar spinal cord compared to midbrain and cortex (Repeated measures ANOVA and Tukey's post hoc: SC vs

midbrain $P = 0.001$, SC vs cortex $P = 0.0021$, midbrain vs cortex $P = 0.149$; number of sections per region = 8 per mouse and $N = 6$) (Fig. 7i). To validate that *mCherry* positive neurons in the lumbar spinal cord were V1 interneurons, RNAscope was performed with probes recognizing *En1* and *mCherry*. Microphotographs in Fig. 7j, k show overlap between the two ($N = 3$). Moreover, a battery of tests was performed to assess potential side effects upon systemic administration in all genotypes between P50 and P70. Observation obtained from the open field test (OFT) (Fig. 7l) show no differences in distance traveled among the genotypes (One-way ANOVA and Dunnett's post hoc: WT $P = 0.0841$, En1$^{cre}$ $P = 0.1674$, SOD1$^{G93A}$;En1$^{cre}$ $P = 0.8586$ $N = 4$−6) (Fig. 7m) nor in time spent in the periphery vs the center of the arena (One way ANOVA and Dunnett's post hoc: WT $P = 0.7487$, En1$^{cre}$ $P = 0.7147$, SOD1$^{G93A}$;En1$^{cre}$ $P = 0.3574$, $N = 4$−6) (One way ANOVA and Dunnett's post hoc: WT $P = 0.7487$, En1$^{cre}$ $P = 0.7153$, SOD1$^{G93A}$;En1$^{cre}$ $P = 0.3579$, $N = 4$−6) (Supplementary Fig. 4e-g). Other cognitive tests assessing memory, anxiety and social behavior were investigated and mice overexpressing *ESYT1* did not show any significant changes at the novel object recognition test (NOR) (One way ANOVA and Dunnett's post hoc: WT $P = 0.9736$, En1$^{cre}$ $P = 0.8236$, SOD1$^{G93A}$;En1$^{cre}$ $P = 0.3020$, $N = 4$−6) (Supplementary Fig. 4a, b); the elevated plus maze test (EPM) (One way ANOVA and Dunnett's post hoc: WT $P = 0.9490$, En1$^{cre}$ $P = 0.7980$, SOD1$^{G93A}$;En1$^{cre}$ $P = 0.4357$, $N = 4$−6) (Supplementary Fig. 4c, d); and the three chamber test (3CT) (One way ANOVA and Dunnett's post hoc: WT $P = 0.8857$, En1$^{cre}$ $P = 0.5389$, SOD1$^{G93A}$;En1$^{cre}$ $P = 0.9967$, $N = 4$−6) (Supplementary Fig. 4h, i). Thus, the systemic administration of h*ESYT1* is safe, shows a limited number of off-targets within the midbrain and the cortex in absence of behavioral side effects.

## ESYT1 systemic administration improves motor functions compared to intraspinal delivery

Recovery of motor performances was also investigated after systemic administration and the three genotypes were compared; WT injected with saline, SOD1$^{G93A}$ injected with AAV-PHP.eB-hSYN-eGFP and SOD1$^{G93A}$;En1$^{cre}$ injected with AAV-PHP.eB-hSYN-DIO-hESYT1. Despite the small N, mice overexpressing *ESYT1* showed dramatic amelioration of motor performances even when compared with the intraspinal injections. Average speed of locomotion was improved from P77 (Two-way ANOVA and Dunnett's post hoc, $P = 0.0222$) (Fig. 8a), step frequency from P63 (Two-way ANOVA and Dunnett's post hoc, $P = 0.0278$) (Fig. 8b), stride length from P56 (Two-way ANOVA and Dunnett's post hoc, $P = 0.0432$) (Fig. 8c), and peak acceleration from P70 (Two-way ANOVA and Dunnett's post hoc, $P = 0.0248$; WT $N = 6$; SOD1$^{G93A}$ $N = 6$, SOD1$^{G93A}$;En1$^{cre}$ $N = 4$; all quantifications were performed in triplicates) (Fig. 8d). At P112 all parameters but peak acceleration were still significantly improved when compared to the SOD1$^{G93A}$ mice (average speed One-way ANOVA and Dunnett's post hoc, SOD1$^{G93A}$;En1$^{cre}$ vs SOD1$^{G93A}$ $P = 4.1e$-10; step frequency One-way ANOVA and Dunnett's post hoc, $P = 5e$-08; stride length One-way ANOVA and Dunnett's post hoc, $P = 0.00903$; peak acceleration One-way ANOVA and Dunnett's post hoc, $P = 0.27007$; WT $N = 6$; SOD1$^{G93A}$ $N = 6$, SOD1$^{G93A}$;En1$^{cre}$ $N = 4$, all quantifications were performed in triplicates) (Fig. 8e–h). Also kinematic analysis showed amelioration of intralimb coordination and reduction of hyperflexion of the hindlimbs with knee angle (One-way ANOVA and Dunnett's post hoc, $P = 1.1e$-05) (Fig. 8j), ankle angle (One-way ANOVA and Dunnett's post hoc, $P = 0.00079$) (Fig. 8k) and foot angle (One-way ANOVA and Dunnett's post hoc, $P = 0.0065$) (Fig. 8l) being significantly improved compared to SOD1$^{G93A}$ mice. Here, knee angle was restored, while the hip angle still remained unchanged (Fig. 8i). Also in this case, SOD1$^{G93A}$;En1$^{cre}$ mice were able to better support their weight against gravity, indicative of increased force (One-way ANOVA and Dunnett's post hoc, $P = 1.5e$-05) (WT $N = 6$; SOD1$^{G93A}$ $N = 6$, SOD1$^{G93A}$;En1$^{cre}$ $N = 4$, all quantifications were performed in

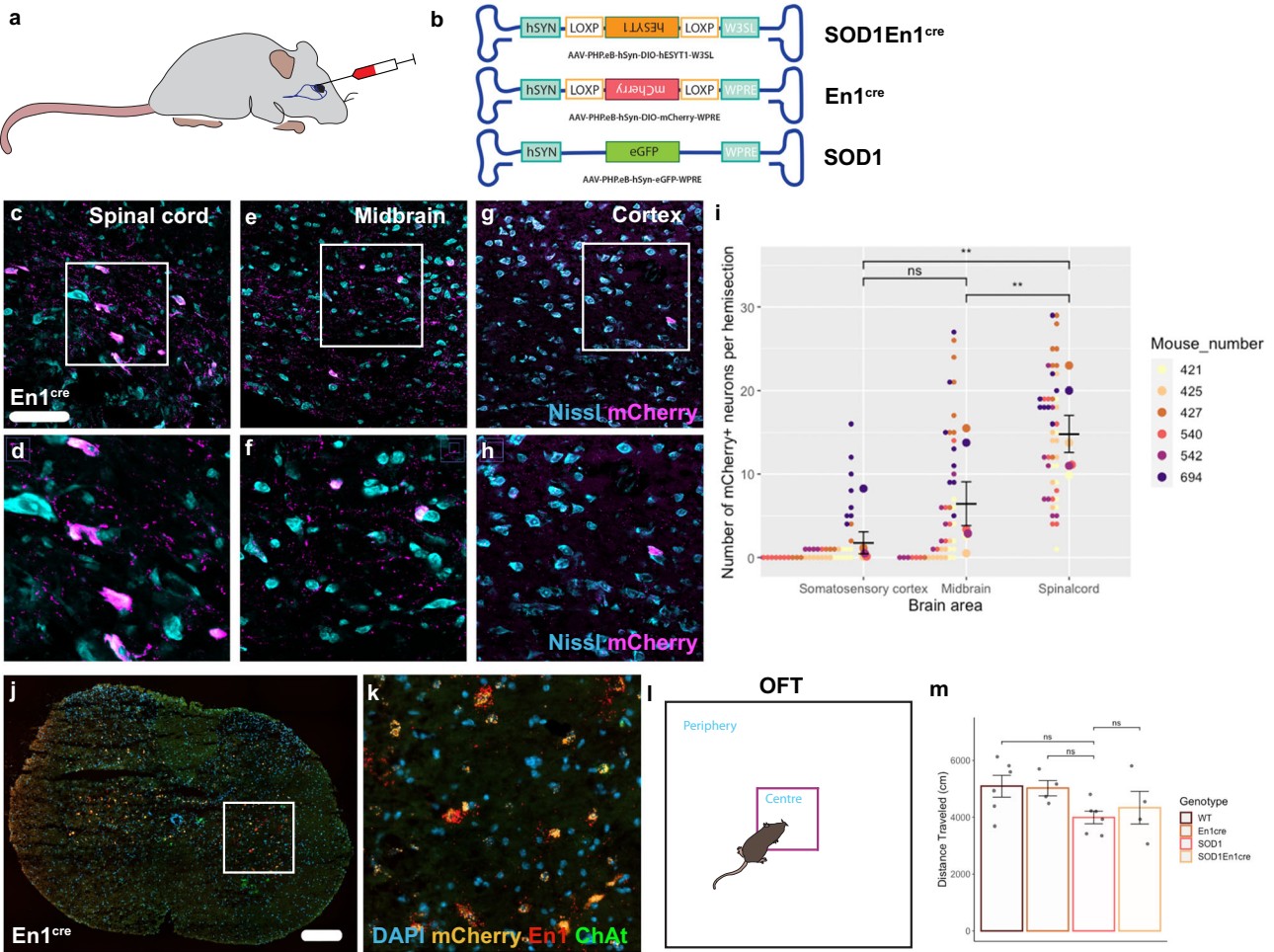

**Fig. 7 | Characterization of systemic overexpression of *ESYT1* obtained by an AAV-PHP.eB vector. a** Cartoon depicting systemic administration via retroorbital intravenous injections of the viral constructs. **b** AAV vectors forcing overexpression of *ESYT1*, *mCherry*, or *eGfp* were injected in SOD1[G93A];En1[cre], En1[cre], and SOD1[G93A] mice respectively at a volume of 100 µl per mouse. **c–h** Microphotograph of spinal cord (**c–d**), midbrain (**e–f**), and cortex (**g–h**) coronal sections of En1[cre] mice showing cre-dependent *mCherry* overexpression upon AAV-PHP.eB-hSYN-DIO-mCherry-W3SL administration. mCherry protein investigation shows successful viral targeting of the lumbar segments of the spinal cord and significantly higher transfection of the spinal cord compared to midbrain and cortex (Nissl in light blue, mCherry in magenta). Higher magnification images are shown in (**d, f, h**). **i** Quantification of mCherry positive neurons, larger dots represent the mean per mouse, while smaller dots represent the number of positive neurons per hemi section (Repeated measures ANOVA and Tukey's post hoc: SC vs midbrain $P = 0.001$, SC vs cortex

$P = 0.0021$, midbrain vs cortex $P = 0.149$, $n = 8$ per mouse and $N = 6$, two-tailed). **j–k** Microphotograph of the *mCherry* overexpression in the ventral horns of the lumbar spinal cord of an En1[cre] mouse upon systemic administration of the viral vector, using RNAscope. *mCherry* expression is specific to *En1* positive neurons, thus selective for V1 interneurons (DAPI in blue, *mCherry* in orange, *En1* in red, *Chat* in green). **l** Schematic depiction of the open field test (OFT), mice were assessed for general locomotor activity with the aid of automated tracking. **m** OFT revealed no significant differences in the distance traveled in any of the experimental conditions suggesting no deleterious effect after systemic delivery (One-way ANOVA and Dunnett's post hoc: WT $P = 0.0841$, En1[cre] $P = 0.1674$, SOD1[G93A];En1[cre] $P = 0.8586$, WT $N = 6$, En1[cre] $N = 4$, SOD1[G93A] $N = 6$, SOD1[G93A];En1[cre] $N = 4$). Scale bar in **c** = 100 µm and in **j** = 200 µm. All graphs show mean values ± SEM. Source data are provided as a Source Data file.

triplicates). Overall, 75% of the SOD1[G93A];En1[cre] mice did not show onset of locomotor phenotype throughout longitudinal assessment (Fig. 9a): percentage shows that between P84 and P98 all SOD1[G93A] mice lose the ability to perform the task, while all SOD1[G93A];En1[cre] but one could maintain 20 cm/s at P112 (SOD1[G93A] median = 87.5, two-tailed Gehan-Breslow-Wilcoxon test, $P = 0.0073$, Chi square= 7.2, df = 1; SOD1[G93A] $N = 6$, SOD1[G93A];En1[cre] $N = 4$). Altogether, these data show a dramatic improvement of motor functions in the SOD1[G93A];En1[cre] mice overexpressing *ESYT1* upon systemic administration.

**Despite improved motor functions and spinal motor neuron survival, SOD1[G93A];En1[cre] do not show increased life span due to loss of weight**

Finally, survival was assessed in the SOD1[G93A];En1[cre] injected with AAV-PHP.eB-hSYN-DIO-h*ESYT1* and the SOD1[G93A] injected with AAV-PHP.eB-

hSYN-eGFP. As mentioned above, SOD1[G93A];En1[cre] mice showed improved motor phenotype throughout longitudinal assessment (Fig. 9a). Moreover, anatomical analysis revealed increased number of spared motor neurons in the lumbar spinal cord (t-test, $P = 0.0074$, $n = 10$ hemicords per mouse SOD1[G93A] $N = 5$ and SOD1[G93A];En1[cre] $N = 4$) (Fig. 9b–e). Although the SOD1[G93A] mice show onset of disease in the lumbar segment of the spinal cord, we investigated motor neuron survival also in the cervical segments of the spinal cord since we expected our treatment to act systemically and V1 interneurons are known to be present also at thoracic and cervical levels of the spinal cord[27]. Here, we found a tendency towards increased number of motor neurons although not significant (t-test, $P = 0.0527$, $n = 10$ hemicords per mouse SOD1[G93A] $N = 5$ and SOD1[G93A];En1[cre] $N = 4$) (Fig. 9d–e). It is worth mentioning that the cervical segment of the spinal cord might be less affected at this timepoint compared to the lumbar. Despite these promising results, SOD1[G93A];En1[cre] life span was not prolonged by

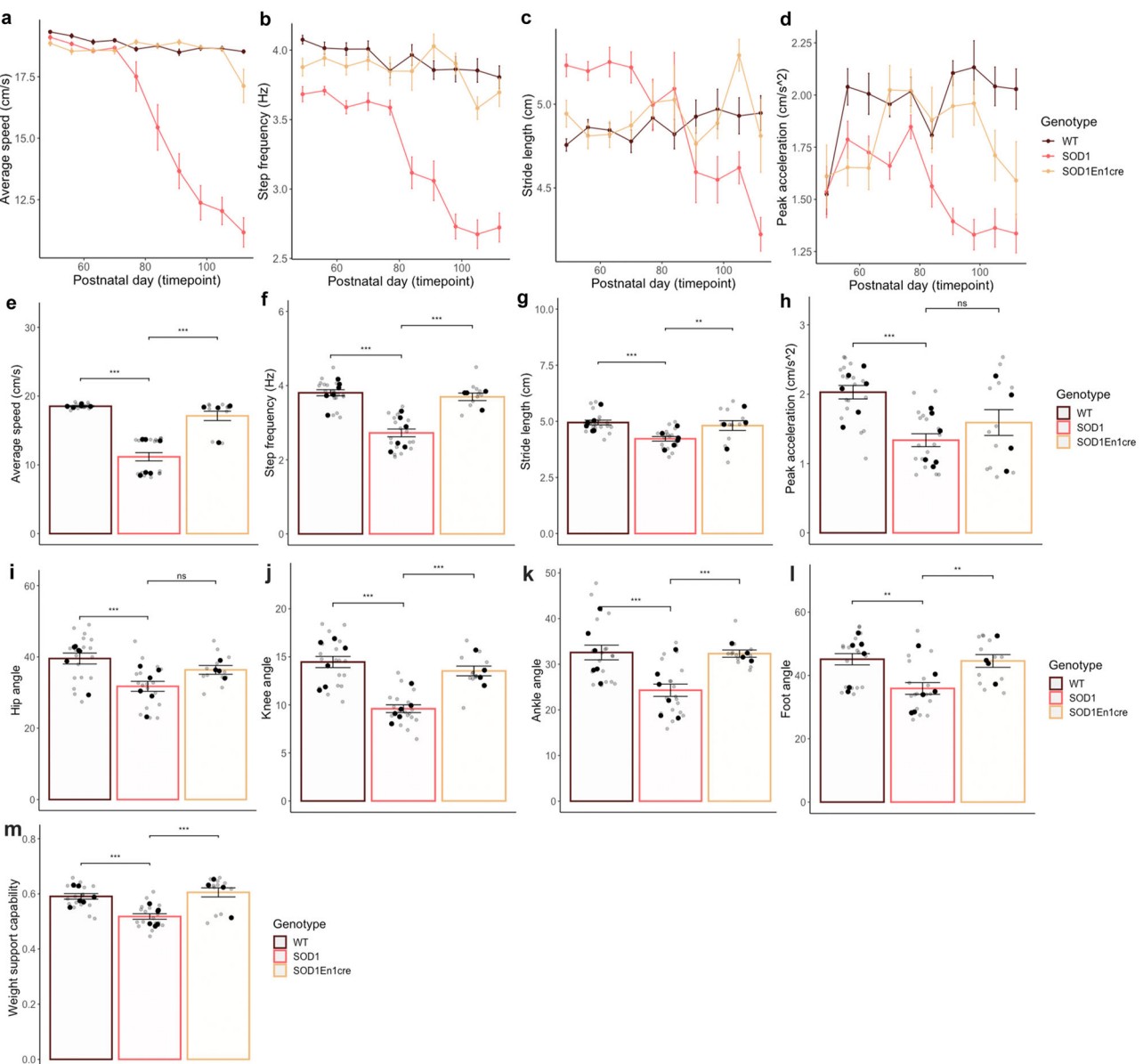

**Fig. 8 | Amelioration of locomotor phenotype in SOD1$^{G93A}$ mice upon systemic *ESYT1* overexpression. a** Average speed analyzed between day 49 and 112 shows amelioration in SOD1$^{G93A}$;En1$^{cre}$ mice from P77 (Two-way ANOVA and Dunnett's post hoc, $P = 0.0222$), (**b**) step frequency from P63 (Two-way ANOVA and Dunnett's post hoc, $P = 0.0278$), (**c**) stride length from P56 (Two-way ANOVA and Dunnett's post hoc, $P = 0.0432$), and (**d**) peak acceleration from P70 (Two-way ANOVA and Dunnett's post hoc, $P = 0.0248$; WT $N = 6$; SOD1$^{G93A}$ $N = 6$, SOD1$^{G93A}$;En1$^{cre}$ $N = 4$; all quantifications were performed in triplicates). At P112 timepoint (**e**) average speed is higher in SOD1$^{G93A}$;En1$^{cre}$ compared to SOD1$^{G93A}$ mice (One-way ANOVA and Dunnett's post hoc, SOD1$^{G93A}$;En1$^{cre}$ vs SOD1$^{G93A}$ $P = 4.1e-10$), as well as (**f**) step frequency (One-way ANOVA and Dunnett's post hoc, $P = 5e-08$) and (**g**) stride length (One-way ANOVA and Dunnett's post hoc, $P = 0.00903$), while (**h**) peak acceleration was not significantly changed at this timepoint (One-way ANOVA and Dunnett's

post hoc, $P = 0.27007$) (WT $N = 6$; SOD1$^{G93A}$ $N = 6$, SOD1$^{G93A}$;En1$^{cre}$ $N = 4$, all quantifications were performed in triplicates). Changes in joint angles were analyzed for (**i**) hip angle, (**j**) knee angle, (**k**) ankle angle and (**l**) foot angle. Significant changes were found for the ankle angle (One-way ANOVA and Dunnett's post hoc, $P = 0.00079$), foot angle (One-way ANOVA and Dunnett's post hoc, $P = 0.0065$), and knee angle (One-way ANOVA and Dunnett's post hoc, $P = 1.1e-05$) upon *ESYT1* overexpression. **m** Weight support was successfully recovered in SOD1$^{G93A}$;En1$^{cre}$ compared to SOD1$^{G93A}$ mice (One-way ANOVA and Dunnett's post hoc, $P = 1.5e-05$) (WT $N = 6$; SOD1$^{G93A}$ $N = 6$, SOD1$^{G93A}$;En1$^{cre}$ $N = 4$, all quantifications were performed in triplicates). All quantifications were performed in triplicates; all graphs show mean values ± SEM. Average values in (**e**–**m**) are shown in black and technical triplicates are shown in gray. Source data are provided as a Source Data file.

*ESYT1* administration (SOD1$^{G93A}$ mean survival 162.7 ± 6, SOD1$^{G93A}$;En1$^{cre}$ mean survival 156.5 ± 9.6, two-tailed Gehan-Breslow-Wilcoxon test, $P = 0.3827$, Chi square = 0.7619, df = 1; SOD1$^{G93A}$ $N = 6$, SOD1$^{G93A}$;En1$^{cre}$ $N = 4$) (Fig. 9g) due to the loss of weight which was comparable to the untreated mice later in disease (Fig. 9f). In conclusion, *ESYT1* systemic overexpression significantly improves motor functions and rescue motor neuron survival until end stage, however, does not prevent weight loss.

## Discussion

Altogether, these results indicate that motor impairment in SOD1$^{G93A}$ mice can be attenuated by stabilization of synaptic inputs between V1 interneurons and motor neurons. By overexpression of ESYT1 presynaptic protein in V1 interneurons, we were able to stabilize inhibitory synaptic connectivity on motor neurons, retain bona fide inhibitory synapses, increase motor neuron survival, and ameliorate motor phenotypes. We cannot exclude that inhibitory interneurons other

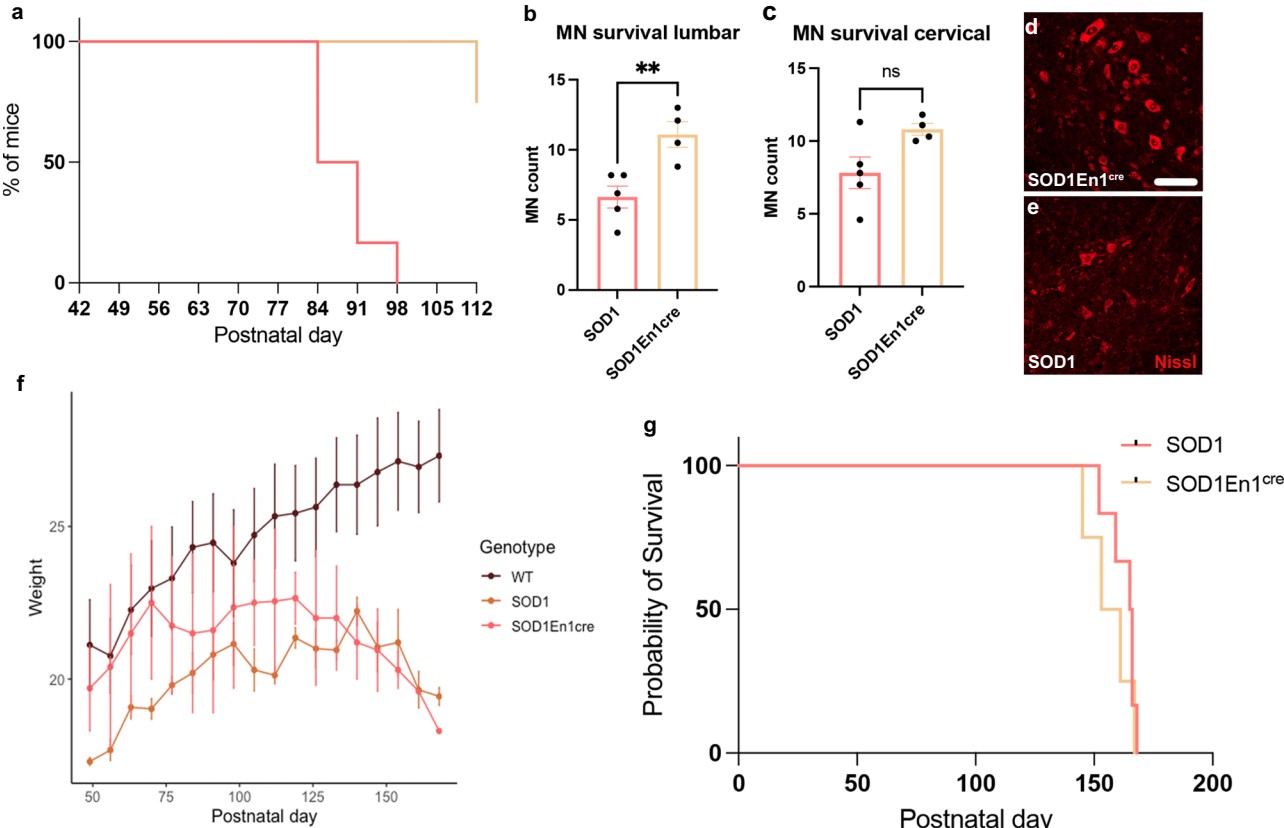

**Fig. 9 | Probability of motor performance, quantifications of spared motor neurons at end stage, weight loss and survival curve. a** Performance of SOD1^G93A;En1^cre and SOD1^G93A mice at forced locomotion assessed on a treadmill at a speed of 20 cm/s. Percentage shows that between P84 and P98 all SOD1^G93A mice lose the ability to perform the task, while all SOD1^G93A;En1^cre but one could maintain 20 cm/s at P112 (SOD1^G93A median = 87.5, two-tailed Gehan–Breslow–Wilcoxon test, $P = 0.0073$, Chi square = 7.2, df = 1; SOD1^G93A $N = 6$, SOD1^G93A;En1^cre $N = 4$). **b** This motor amelioration is paralleled by increased retention of motor neurons in the lumbar spinal cord (t-test, two-tailed, $P = 0.0074$, $n = 10$ hemi cords, SOD1^G93A $N = 5$; SOD1^G93A;En1^cre $N = 4$). **c** Motor neuron quantifications at cervical level of the spinal cord in the SOD1^G93A;En1^cre compared to the SOD1^G93A mice(t-test, two-tailed, P = 0.0527, $n = 10$ hemi cords, SOD1^G93A $N = 5$; SOD1^G93A;En1^cre $N = 4$). Micro-photographs show motor neuron survival in the lumbar spinal cord for (**d**)

SOD1^G93A;En1^cre and (**e**) SOD1^G93A mice. **f** Weight loss during experimental course. No significant differences were observed between SOD1^G93A;En1^cre and SOD1^G93A mice (From P49 to P140: WT $N = 6$, SOD1^G93A $N = 6$, SOD1^G93A;En1^cre $N = 4$). Then: P147: WT $N = 6$, SOD1^G93A $N = 6$, SOD1^G93A;En1^cre $N = 3$, P154: WT $N = 6$, SOD1^G93A $N = 6$, SOD1^G93A;En1^cre $N = 3$, P161: WT $N = 6$, SOD1^G93A $N = 5$, SOD1^G93A;En1^cre $N = 2$, P168: WT $N = 6$, SOD1^G93A $N = 4$, SOD1^G93A;En1^cre $N = 1$). Since males and females were used at similar ratio in all conditions the graph includes weights of both genders. **g** Survival curve of SOD1^G93A;En1^cre and SOD1^G93A mice. No significant differences in survival rates were observed (two-tailed Gehan–Breslow–Wilcoxon test, $P = 0.3827$, Chi square = 0.7619, df = 1; SOD1^G93A $N = 6$, SOD1^G93A;En1^cre $N = 4$). Scale bar in (**d**, **e**) = 50 µm. All graphs show mean values ± SEM. Source data are provided as a Source Data file.

than V1 may also be affected at the symptomatic timepoints investigated in the present study and may contribute to the overall Vgat+ synaptic loss observed at P112 (Fig. 3c). We now know that at P112 spinal network degeneration is further exacerbated and also excitatory V2a interneurons appear affected in the SOD1^G93A mice[28]. However, future investigations will be required to reveal the fate of other inhibitory interneurons later in disease (including V2b and V0d populations). Our present results suggest that presymptomatic V1 restricted-*ESYT1* overexpression might also have an effect at network level, since Vgat inputs are comparable to control conditions after treatment. In fact, *ESYT1* might be able to stabilize connectivity between V1 interneurons and other targets as well, not only motor neurons.

Both viral approaches e.g., intraspinal and systemic administrations, significantly improved motor functions in the treated SOD1^G93A mice, with the latter reducing 75 % of the onset of motor impairment in the ALS mice. Here, mice showed increased average speed, step frequency and stride length, as well as peak acceleration during forced locomotion. Importantly, *ESYT1* overexpression also partially recovered intralimb coordination of the hindlimbs, with restoration of the knee, ankle, and foot angles. All these being V1-dependent phenotypes[2,6,29]

which are corrected in the SOD1^G93A; En1^cre mice. Not only did mice show improved gait and intralimb kinematics upon treatment, but they were also able to better support their bodies during locomotion, which is indicative of retained force in their hindlimbs, allowing them to counteract gravity. Despite the striking amelioration of motor functions and the increased spinal motor neuron survival at end stage, we did not observe an increased life span after *ESYT1* systemic administration due to body weight loss. While this is not the hoped outcome, we believe that these negative results highlight the importance of other neural circuits affected in disease that might play a pivotal role and should be considered when interpreting our results.

Previous studies have shown the presence of V1 interneurons at thoracic and cervical levels[27], however it remains unknown if these interneurons are also present in the rhythmogenic circuits controlling chewing, swallowing and respiration[30]. Our systemic administration aimed to potentially target also these neural networks, which, as well as the central pattern generators in the lumbar spinal cord, coordinate rhythmic activity of neurons[30]. Recent results reported that mesencephalic trigeminal neurons controlling masticatory rhythm modulation show electrophysiological abnormalities already at 12 weeks (P84) in the SOD1^G93A mice which correlates with simultaneous decrease in

body weight[31]. In the SOD1$^{G93A}$;En1$^{cre}$ mice overexpressing *ESYT1* we observe a clear decrease in body weight at this same timepoint. Moreover, melanin-concentrating hormone (MCH)-positive neurons found in the lateral hypothalamic area and controlling metabolic homeostasis degenerate in ALS[32]. In addition, the orbitofrontal-hypothalamic projections were found disrupted in the symptomatic SOD1$^{G93A}$ mice, thus suggesting their contribution to the hypermetabolic phenotype[33]. Due to the V1-restricted overexpression of *ESYT1* it is unlikely that our treatment could act on these neurons. Thus, these studies, together with our results, point towards the need for understanding how specific neural circuits degenerate in ALS to find future combinatorial therapies. Previous studies have shown that chemogenetic activation of the V2a excitatory interneurons found in the brainstem and spinal cord alone is sufficient to increase accessory respiratory muscle activity and enhance ventilation in mice[34]. It has been previously hypothesized that therapies targeting specific microcircuit dysfunctions might help slowing down the course of the disease[35]. The present work supports this hypothesis and indicates that the V1 interneurons-motor neuron circuit can be a potential target for treatment of spinal motor dysfunctions in ALS.

Our future directions will aim to increase the translational applicability of our treatment. Electrophysiological studies investigating recurrent inhibition support general inhibition dysregulation in ALS patients, and reduced inhibition in patients showing initial motor weakness in the lower limbs[36]. This could be used in combination with the changes in gait observed early on in ALS patients with both spinal and bulbar onsets including reduction in stride length and instability during complex walking situations[37]. Thus, further effort will be required to identify a potential therapeutic window for targeting premotor circuits at the appropriate timepoint to improve motor functions. Furthermore, the surprising maladaptive changes observed in healthy En1$^{cre}$ mice upon *ESYT1* overexpression, leading to motor neuron shrinkage, suggest the need for targeted administration of this treatment. Extended synaptotagmins not only play a role in synaptic growth and stabilization[16], but thanks to their C2 domain, they also act on membrane tethering. Both these events are regulated by Ca$^{2+}$[38]. Esyt1 facilitates binding of the endoplasmic reticulum (ER) and the plasma membrane, while inducing lipid transports between them in a Ca$^{2+}$-dependent manner. Thus, Esyt1 controls also lipid homeostasis[39]. Moreover, Esyt1 is also involved in the mitochondria-ER interactions, hence contributing to the general cellular homeostasis[40]. Previous studies have shown that Esyt1 is highly expressed in the ventral horn of the spinal cord in healthy controls[41], for this reason it is difficult to anticipate how other healthy neurons would react to its overexpression. Esyt1 disruption is known to lead to defective Ca$^{2+}$ signaling[42]. Furthermore, Vamp proteins (also contributing to ER membrane tethering), lipid transport, neurotransmitter release, and stabilization of presynaptic microtubules were suggested to be involved in motor neuron disease[43]. For these reasons, further investigations are needed to elucidate motor neuron shrinkage after *ESYT1* overexpression in healthy V1 interneurons. Moreover, *En1* is expressed in neuronal populations outside the spinal cord (e.g., dopaminergic neurons) that could be affected by *ESYT1* overexpression. Hence, the use of specific enhancers[44] or synthetic synaptic organizers[45] could be viable alternatives for selective targeting of affected circuits.

## Methods

### Ethical permits and mouse strains
All experiments comply with the relevant ethical regulations of the EU Directive 20110/63/EU and approved by the Danish Animal Inspectorate (Ethical permits: 2018-15-0201-01426 and 2022-15-0201-01164, University of Copenhagen). SOD1$^{G93A}$ (B6.Cg-Tg(SOD1-G93A)1Gur/J) stock no: #004435 were retrieved from Jackson Laboratory, while En1$^{cre}$ mice, kept on a C57BL6/J background, were provided by Assistant Prof. Jay Bikoff (St. Jude Children's hospital, St Louis, Texas USA).

Mice were genotyped using DNA extracted from earclipping and the following primers were used: for the SOD1$^{G93A}$ gene, 5'-CAT CAG CCC TAA TCC ATC TGA-3' and 5'-CGC GAC TAA CAA TCA AAG TGA-3', while for the En1$^{cre}$ gene, 5'-GAG ATT TGC TCC ACC AGA GC-3' and 5'-AGG CAA ATT TTG GTG TAC GG-3'. Copy number of the mutated SOD1 gene was quantified by qPCR utilizing the primers 5'-GGG AAG CTG TTG TCC CAA G-3' and 5'-CAA GGG GAG GTA AAA GAG AGC-3' for the SOD1$^{G93A}$ gene. The qPCR reaction was performed as suggested by the mice supplier. All mouse strains were bred with congenic C57BL6/J mice, stock no: #000664 (Jackson Laboratory). Mice were housed according to standard conditions with ad libitum feeding, constant access to water, and a 12:12 h light/dark cycle. All mice, including multiple crossing, were genotyped, tested for copy number, and phenotype was assessed, including weekly weight. For survival experiments, the humane endpoint was defined as a weight loss of 15 % and/or functional paralysis in both hind limbs together with inability to perform a righting test < 20 s. Mice of both genders were included in the study and non-transgenic SOD1$^{G93A}$ littermates were used as controls. The detailed count of males (M) and females (F) per experiment, listed for each figure, is as follows: Fig. 1d: M 14, F 20; Fig. 1g: M 11, F 16; Fig. 2k: M 0, F 8; Fig. 2x: M 6, F 6; Fig. 3e: M 9, F 9; Fig. 3n: M 9, F 11; Fig. 3o: M 5, F 4; Fig. 5u, v: M 5, F 5; Fig. 6b–e: M 19, F 16; Fig. 6f–q: M 16, F 16; Fig. 7i: M 3, F 3; Fig. 7m: M 7, F 13; Fig. 8a–m: M 6, F 10; Fig. 9a: M 4, F 6; Fig. 9b, c: M 4, F 5; Fig. 9f: M 6, F 10; Fig. 9g: M 4, F 6; S Fig. 1a: M 13, F 7; S Fig. 1b: M 14, F 0; S Fig. 1c: M 9, F 7; S Fig. 2a: M 19, F 16; S Fig. 2c–f: M 19, F 16; S Fig. 3e–t: M 3, F 1; S Fig. 4b: M 7, F 12; S Fig. 4d–i: M 7, F 13.

### Adeno-associated Viral Vector production for intraspinal injections
The AAV overexpressing *ESYT1* was developed utilizing a pAAV-hSYN-DIO[hCAR]off-[hM4Di-mCherry]on-W3SL backbone construct containing the W3SL cassette (Addgene plasmid #111397). First, hCAR and hM4di-mCherry were exerted by double digestion utilizing *Asc1/Nhe1* restriction enzymes, the *mCherry* fluorescent tag was exerted, and Kozak sequences inserted by PCR. cDNA of the human Extended Synaptotagmin 1 (h*ESYT1*) was purchased by Dharmacon (#MHS6278-202826307) and insert amplified by PCR utilizing Primers 5'-TAG CAG GCG CGC CCT AGG AGC TGC CCT TGT CC-3' and 5'-GAG TCT CTA GAG CCA CCA TGG AGC GAT CTC AGG AG AG-3' containing a Kozak sequences at the start codon position. The PCR product was double digested utilizing *Xbal/AscI* restriction enzymes and ligated into the pAAV backbone. Both backbone construct and insert were sequenced to validate successful sequence-content and orientation. The pAAV-vector, encoding for *ESYT1*, the pHelper carrying the adenovirus-derived genes, and pAAV-Rep-Cap (carrying the AAV2 replication and AAV8 capsid genes) were co-transfected in 293-cells with Dulbecco's Modified Eagle Medium (DMEM) supplemented with 10% fetal bovine serum (FBS) and 1 % penicillin-streptomycin (all from GeneMedi). The following day, medium was replaced. Medium containing viral particles was harvested ~72 h later through low-speed centrifugation at 1.500 g in an Eppendorf centrifuge (Eppendorf 5810 R, Hamburg, Germany) for 5 min. Cell pellet was resuspended in 10 mM tris(hydroxymethyl)aminomethane hydrochloride (pH 8,5) lysis buffer with subsequent freeze/thaw cycles. Supernatant containing the harvested AAV-particles was collected after 10 min centrifugation at 3.000 g and filtered through a 0,22 μm filter. Concentration of the filtered supernatant occurred by ultrafiltration and several centrifugation steps. The pellet containing the viral particle was resuspended in 0,1 M PBS pH 7,4 (#10388739, Gibco) and aliquoted and stored at −80 °C until further use. The AAV8-hSYN-DIO-mCherry vector was purchased from Addgene (plasmid #50459: Bryan Roth lab). The two viral vectors had equal titers of 4.0 × 10$^{12}$ vg/ml.

### Intraspinal injections and viral delivery
To achieve cre-dependent overexpression of h*ESYT1* in V1 interneurons, En1$^{cre}$ mice were used and crossed with SOD1$^{G93A}$. For

anatomical and behavioral experiments performed in this study, all littermates (including WT, SOD1$^{G93A}$;En1$^{cre}$, En1$^{cre}$, and SOD1$^{G93A}$ mice) were injected with AAV8-hSYN-DIO-hESYT1-W3SL at postnatal day 30 by an experimenter blind to the genotype. The four injected genotypes were included in the study, to investigate off target effects of the viral vector and changes due to the surgical procedure. For intraspinal injections mice were anaesthetized with 2% isoflurane and the lumbar level of the spinal cord was exposed for stereotaxic injections (Neurostar). A small incision was performed with micro-scissors between the vertebrae to deliver the virus in the L1, L2 and L3 spinal segments. For visualization, virus was mixed with 4% fast green (#A16520.22, Invitrogen) dissolved in saline and injected using a glass micropipette at a rate of 100 nl/min. The micropipette was kept in place for 2 min after viral delivery to avoid backflow. Bilateral injections of 100 μl AAV8-hSYN-DIO-hESYT1-W3SL virus were performed. Additionally, similar injections of 100 μl of a control AAV8-hSYN-DIO-mCherry-WPRE were conducted to validate successful cre-recombinase upon injection. Pre-operatively mice were treated with a subcutaneous injection of buprenorphine diluted in saline at a concentration of 0.3 mg/ml. Moreover, post-operatively pain relief was administered for 3 days using a similar dose of buprenorphine (Temgesic) mixed in DietGel Boost (ClearH$_2$O). For dual viral injections AAV8-hSYN-DIO-hESYT1-W3SL virus was mixed at a concentration of 1:1 with the AAV-phSYN1(S)-FLEX-tdTomato-T2A-SypEGFP-WPRE (titer $5.56 \times 10^{11}$/ml, Viral Vector Core, Salk Institute for Biological Sciences; Addgene #51509) and a final volume of 100 nl was injected in spinal segments L1-L3 as described above. Three En1$^{cre}$ and three SOD1$^{G93A}$;En1$^{cre}$ were injected and tissue was collected 3 weeks after viral delivery.

## Adeno-associated Viral Vector production for intravenous injections

AAV-PHP.eB-hSYN-DIO-hESYT1-W3SL, AAV-PHP.eB-hSYN-DIO-mCherry-W3SL and AAV-PHP.eB-hSYN-GFP-W3SL viral expression was under control of the human Synapsin (SYN) promoter, which confers neuron-specific, long-term expression[46]. The PHP.eB capsid was used to increase transduction efficiency in central nervous system and blood brain barrier crossing ability since the virus was intravenously administered[26]. The viruses were produced by the Verhaagen lab at Netherlands Institute for Neuroscience. Viral production protocol uses cells in suspension, rather than monolayer plated cells, which enables production of a titer in the $10^{11}$ range.

## Intravenous injections and viral delivery

Three AAV-PHP.eBs were used for systemic administration experiments. AAV-PHP.eB-hSYN-DIO-hESYT1-W3SL was employed in the SOD1$^{G93A}$En1$^{cre}$ mice to overexpress ESYT1 in presence of cre-recombinase. AAV-PHP.eB-hSYN-DIO-mCherry-W3SL was employed in the En1$^{cre}$ mice to assess the spread of the virus upon cre-recombinase. AAV-PHP.eB-hSYN-GFP-W3SL was employed in the SOD1$^{G93A}$ mice as controls. Saline injections were used for the WT control mice which act as sham treatment. All viruses were injected at a volume of 100 μL per mouse, and a titer of $3.03 \times 10^{11}$ viral genomes (vg) per 100uL was used. The respective viruses and saline were injected between P28 and P35. To obtain systemic spread of the virus, retro-orbital intravenous (IV) injections were delivered under general anesthesia with isoflurane (4% in 100% oxygen for induction, 1.5–2% in 100% oxygen for maintenance). As mCherry signal fades over time, the En1$^{cre}$ mice were euthanised at P70, and tissue collected for anatomical analysis of viral transduction via either immunohistochemistry or RNAscope.

## RNAscope protocol

For RNAscope assay, mice were anaesthetized with an overdose of Pentobarbital (250 mg/kg) and sacrificed by decapitation. Spinal cords were dissected, snap frozen in isopentane (2-methylbutane, Uvasol, Merck) and kept in dry ice for cryoprotection. Then, spinal cords were coronally or longitudinally sectioned at 12 μm-thickness by using a cryostat (NX50, Epredia), collected on Superfrost Plus slides (#12312148, Epredia) and stored at −80 °C until further processing. Samples were pre-treated and processed using the RNAscope Multiplex Fluorescent v2 Assay following the supplier's protocol (Advanced Cell Diagnostics – ACD). Briefly, sections were fixed in 4% PFA (#HL96753, HistoLab) for 15 min at room temperature, washed in DPBS and dehydrated with sequential ethanol steps. Then, they were incubated with RNAscope hydrogen peroxide solution for 10 min at room temperature, rinsed in DEPC-treated water, treated with RNAscope protease IV for 30 min at room temperature, and washed in PBS before in situ hybridization. Probes were hybridized for 2 h at 40 °C in HybEZ oven (#PN 321710, ACD Bio), samples were stored in 5 X Saline Sodium Citrate (SSC) overnight at room temperature, followed by incubation with signal amplification and developing reagents according to the manufacturer's instructions. Probes were purchased from ACD: Hs-ESYT1-C1 (catalog #540391), pAAV-hESYT1-WPRE-O1-C3 customized probe (catalog #1062751-C3), Mm-En1-C1 (catalog #442651), Mm-Chat-C2 (catalog #408731-C2), tdTomato-C3 (catalog #317041-C3). The hybridized probe signal was visualized and captured on a Zeiss LSM 900 confocal microscope with a Plan-Apochromat 40 x oil objective – NA = 1,4 or 20 x air objective – NA = 0,8, zoom = 1. Image analysis of RNAscope microphotographs was performed on tiled images of the ventral region of each section. A total of 10 tiled images were analysed per mouse and condition. Quantification of ESYT1 positive cells was performed manually by an experimenter blind by the condition and validated with the open-sourced bio-image analysis software QuPath (version 0.2.3)[47], using the parameters shown in Supplementary Table 1. For cell segmentation NeuroTrace 640 or 435 (1:200, #N21483, #N21479 Invitrogen) was used instead of DAPI. Quantifications are presented as total amount of positive cells/total cells (Hs-ESYT1-C1 probe) or total amount of double positive cells/total amount of En1 positive cells (Hs-ESYT1-C1 probe and pAAV-hESYT1-WPRE-O1-C3 probe).

## Immunohistochemistry

Mice were injected with Pentobarbital (250 mg/kg) and transcardially perfused with pre-chilled phosphate buffered saline (PBS, #10388739, Gibco) followed by pre-chilled 4% paraformaldehyde (PFA, #HL96753 HistoLab). Spinal cord was dissected, postfixed for 60 min in cold 4% PFA and cryoprotected in PBS 30% sucrose for 48 h at 4 °C. For immunofluorescence coronal and longitudinal sections of the lumbar spinal cord were sectioned at 30 μm-thickness. Sections were collected on Superfrost Plus slides (#12312148, Epredia) and stored at −20 °C for further processing. Slides were then washed for 10 min in PBS at room temperature and blocked for 1 h in PBS with 0.1% Triton-X100 (PBS-T, Sigma Aldrich) and 1.5% donkey serum (Invitrogen). Sections were incubated for 24–48 h at 4 °C in primary antibodies diluted in blocking solution. The following antibodies were used: DS Red (1:1000, Rabbit, #632496 Takara-Clontech), Chat (1:300, Goat, #AB144P Millipore), Vgat (1:500, Rabbit, #PA5-27569 Millipore), Esyt1 (1:100, Rabbit, #HPA0168589 Sigma), Gephyrin (1:200, Guinea Pig, #147-318-SY Synaptic Systems). Slides were washed three times in PBS at room temperature for 10 min each and then incubated for 1 h with appropriate secondary antibodies (1:500, Alexa Fluor 488, 568, Invitrogen) diluted in blocking solution. Subsequently, another three 10-minute PBS washes, and counterstaining was performed with NeuroTrace 640 (1:200, Invitrogen) or Hoechst (1:2000, Invitrogen). After two 5 min washes in PBS, slides were dried, and cover slipped using ProLong Diamond Antifade Mountant (Invitrogen). Microphotographs were obtained utilizing either Zeiss LSM 700 or 900 confocal microscopes, using a Plan-Apochromat 20 X objective – NA = 0,8, zoom = 0,5. For motor neuron survival and synaptic density quantifications, one image of the ventral region of each hemisection was acquired. Quantifications were conducted utilizing Fiji software. Motor neurons were

detected and quantified based on localization, shape, and cell diameter of 28 μm. For synaptic density quantifications, microphotographs acquired with a 20 x objective (zoom = 1), were transformed to a grayscale 8-bit images and quantified after applying a threshold for signal intensity/background correction. Images with a higher background/noise ratio were excluded. ROI were drawn around the motor neurons of interest that showed clear soma staining and visible nucleus. Pixel contained in the area of interest were quantified with the Fiji function Analyze particles and masks of all quantified images were created to validate the analysis. Integrated intensity of pixels obtained with Fiji was normalized by the motor neuron area.

## Proximity ligation assay

Three weeks after dual viral injections, mice were injected with Pentobarbital (250 mg/kg) and transcardially perfused with pre-chilled phosphate buffered saline (PBS, Gibco) followed by pre-chilled 4% paraformaldehyde (PFA, HistoLab). Spinal cord was dissected, and cryoprotected as described above. Spinal cords were sectioned at 30 μm-thickness. Proximity ligation assay kit was purchased from Merck (DUOLINK in Situ Red Starter kit Goat/Rabbit, #DUO92105) and performed following supplier recommendations. For PLA two antibodies were used the Esyt1 (1:200, Rabbit, #HPA0168589 Sigma), and the Gfp (1:1000, Goat, #PA5-143588, Invitrogen). Antibodies were first validated on tissue separately by immunohistochemistry as described above, and then incubated over night before developing the ligation assay. Negative controls were included for all injected tissue with primary antibodies excluded. Microphotographs were obtained using a Zeiss LSM 900 confocal microscope and Airy scan to increase image resolution, using a 63X oil objective. Experiments were performed in triplicates per condition.

## DigiGait treadmill test

Locomotor performance was assessed using the DigiGait motorized transparent treadmill (Mouse Specifics, Inc.), which allows recording of mice from a ventral and lateral view. Mice were trained 1 week prior, then tested weekly from P49 until P112 on the treadmill at a speed of 20 cm/s for analysis of disease progression. After a 2 min acclimatization to the treadmill, mice were recorded at a speed of 20 cm/s for 10 s in 3 consecutive trials with 2 min rest periods between recordings. Belt speed was adjusted to 15, 10 or 5 cm/s as needed, depending on the locomotor capability.

## Kinematic analysis

The videos captured during the treadmill experiments were analysed utilizing DeepLabCut[25] (DLC) software and an optimized model of the one described in Allodi et al. [2], available at https://doi.org/10.5281/zenodo.10956898. Briefly, eleven digital markers were placed on the mice and the tracks obtained by DLC were further analysed to extract the locomotor measures reported in the article. The reported measures are speed, acceleration (described as sudden increase or decrease in the speed), coordination (used to estimate the stride length and step frequency). The lateral view videos simultaneously captured during the treadmill experiments were analysed with a second DeepLabCut model. Six image markers (corresponding to iliac crest, hip, knee, ankle, foot, and toe) on the left hindlimb were tracked using this model. The tracked positions of the markers from each frame were further analysed to obtain four joint angles (at the hip, knee, ankle, and foot) within each step cycle, as shown in Fig. 6 (panels i-n). The angles reported in the figures are calculated from the difference between the maximum and the minimum inflection angle per joint within a step cycle. This difference angle is then averaged over 15 step cycles per mouse.

## Weight support analysis

The gait analysis tool was implemented with an algorithm for quantification of weight support ability. Three key point markers for the front paw, top of the back, and tail base were placed using DeepLabCut (Fig. 6o). The algorithm calculates a measure of weight support ability based on the distance of the tail base from the treadmill belt, defined as lowest front paw y-coordinate within each step cycle. This was implemented to account for differences across videos. This measure was calculated as mean tail base height as a ratio between the treadmill floor (0) and the top of the animal's back (1), to account for mouse size in the calculation.

$$R = \frac{1}{T} \sum_{t=1}^{T} \frac{y_t - f}{b - f} \qquad (1)$$

Here, R is the tail base ratio as a value between 1 (back) and 0 (floor), $y_t$ is the y-coordinate for the tail base marker for each frame ($t$), $T$ is the total number of frames in each video, f is the mean of the y-coordinates for the front paw marker found at the lowest position within each step cycle, and b is the mean of the y-coordinates for the back marker across all frames. The code was written in Python version 3.9.7 using Jupyter Notebooks and the packages pandas, numpy, and matplotlib.

## Cognitive assessment after systemic administration

A battery of behavioral paradigms was used to investigate potential side effects upon systemic administration across the four genotypes. The behavioral paradigms were conducted in an order of Open Field test and Novel Object Recognition within the same day. Then the Elevated Plus Maze was conducted, followed by the Three-Chamber Test, each on separate days. All paradigms were carried out at least 3 weeks post-injection to allow sufficient time for protein expression, thus between P50 and P70. Mice were acclimatized for 30 minutes before testing. Behavioral tracking of nose, body, and tail base was performed using EthoVision XT software version 17 (Noldus) and a camera mounted above the arenas. EthoVision was also used for automated calculation of relevant parameters of interest, which were all based on tracking of the center point of the animal as it provided the most stable tracking across paradigms. All zones of interest were drawn in EthoVision. The Open Field Test (OFT) was used to test overall locomotor activity. Mice were placed in a 50 cm × 50 cm Plexiglas arena (Fig. 7i) and allowed to explore freely for 10 minutes. Distance moved (cm) and time spent in the periphery and in the center of the arena was analyzed. The Novel Object Recognition (NOR) task was used to evaluate learning and memory. Mice were placed in an arena containing two round contact zones with a radius of approximately one mouse body length. The novel and familiar objects were similar in volume but differed in their appearance and shape. EthoVision tracked time spent in the contact zones with the novel and the familiar object for the calculation of memory performance. The Elevated Plus Maze (EPM) was used to assess changes in anxiety. Mice were placed in the central zone and explored the areas freely for 10 min. EthoVision acquired entries in each of the arms of the maze. More entries in open arms over closed arms reflect anti-anxiety behavior. The three-chamber test (3CT) was used to assess interest in social novelty. The arena was designed to have three chambers, and two relevant zones were established where familial and novel mice would be introduced. The paradigm consisted of three phases: habituation, sociability phase, and social novelty phase. The time spent in each chamber was acquired by EthoVision. Male subjects were presented with other male baits, and female subjects were presented with other female baits.

## Statistics and Reproducibility

*Esyt1* quantification, Esyt1 overexpression, motor neuron quantification (MNq), synaptic density data, and MN size were analyzed by a One-way Analysis of variance (ANOVA) with number of positive cells/total cells (*Esyt1* quantification), number of double positive cells/total amount of *En1* positive cells (*ESYT1* overexpression), number of

positive cells (MNq), and Integrated density/microns (Synaptic density) and motor neuron size (MN size), as dependent variable, and Strain (wt, SOD1$^{G93A}$, SOD1$^{G93A}$; En1$^{cre}$, En1) (ESYT1 overexpression, MNq and SD) or condition (En1$^{cre}$/Inj; wt/Inj) (Esyt1 quantification) (En1$^{cre}$; En1$^{cre}$/Inj) (MN size) as between-groups factor. When appropriate, post hoc comparisons with Dunnett's test correction were performed. Moreover, statistical significance was set at $P < 0.05$, and effect size is reported when appropriate in figure legends: Partial eta-squared values are reported and considered as small (0.01), medium (0.06), or large (0.14). All analyses were performed using JASP$^{©}$ software (JASP Team, University of Amsterdam, version 0.16). For comparing mCherry expression across CNS areas (somatosensory cortex, midbrain and spinal cord), repeated measures ANOVA with Tukey's post-hoc test was employed with mean number of mCherry+ neurons per hemisection for each mouse as the dependent variable, CNS area as the between-groups factor, and mouse number as the within-groups factor. For comparison of locomotor phenotype onset-curves, Log-rank (Mantel-Cox and Gehan-Breslow) test was employed. One-way ANOVA with subsequent post-hoc Dunnet's or Fisher's LSD tests were used to compare differences between genotypes at single timepoints. The genotypes were compared against SOD1 as the reference group, to compare intervention and controls against the diseased mice. When considering multiple variables (genotype and time), two-way ANOVA was used. Two-way ANOVA tests were followed by Dunnett's test. The same approach was used for both ventral and lateral video data. Statistical analysis of behavior was performed in R Studio utilizing packages pacman, tidyverse, DescTools, ggsignif, emmeans, multcomp, broom, readxl, ggbeeswarm, and afex. No statistical method was used to predetermine sample size; our sample sizes are similar to those reported in previous publications[2]. Data collection was performed blind to the conditions of the experiments although, in behavioral assessment, some ALS mice could be recognized at later stages. One mouse was excluded from the study since it showed aberrant locomotor phenotype upon intraspinal injection, all other injected mice were included in the study. All experiments were performed at least in triplicates (except for the experiments were otherwise stated in the Results and Figure legend paragraphs) and observation showed reproducibility among replications. No data were excluded from the analyses. All results are expressed as mean ± SEM, reported n values represent distinct biological replicates. $P < 0.05$ was considered statistically significant. Asterisks in figures and figure legends are *$P < 0.05$, **$P < 0.01$, ***$P < 0.001$, or ****$P < 0.0001$.

### Reporting summary
Further information on research design is available in the Nature Portfolio Reporting Summary linked to this article.

## Data availability
The processed data are available at PURE repository at University of St Andrews https://doi.org/10.17630/940f2947-ae2b-4e18-9493-bd65612ba3a6. Due to the dimension of the files, the videos can be obtained from the corresponding author upon request. Source data are provided with this paper.

## Code availability
The code used to analyze data, produce figure content and videos is available at Zenodo: https://doi.org/10.5281/zenodo.10956898.

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

## Acknowledgements

We thank Prof. Ole Kiehn, Department of Neuroscience – University of Copenhagen, for the access to the surgery room and the use of the DigiGait treadmill. We acknowledge the Core Facility for Integrated Microscopy (CFIM), at the Faculty of Health and Medical Science of University of Copenhagen and the Department of Experimental Medicine (AEM), especially Dr. Pablo Hernandez-Varas (CFIM) and Alex Soelberg Laugesen (AEM). This work was supported by the Lundbeck Foundation (R346-2020-2025, I.A.), the Louis-Hansen Foundation (21-2B-9477/L102, I.A. and 18-2B-3570, R.M.R.), the Danish Society for ALS (1214371001, I.A.), the Laege Sofus Carl Emil Friis og hustru Olga Doris Friis' foundation (1218471001, I.A.), the Danish Society for Neuroscience (DSFN2020-5, R.F. and DSFN2022-3, A.S.), a grant from the research fund of the Royal Netherlands Academy for Arts and Sciences (KNAW - koninklijke academie van Wetenschappen, BDO1095, J.V.), the Department of Neuroscience at University of Copenhagen (I.A.), the School of Psychology and Neuroscience at University of St Andrews (I.A.) and the St Leonard's college PhD Scholarships (W7, I.A.).

## Author contributions

Conceptualization I.A., Methodology I.A., S.M., R.F., A.S., R.M.R, R.S., Viral vector production A.T.S, G.N.H., I.A., R.F., K.P., J.V. Data Analysis S.M., A.S., R.F., H.W., Supervision and Funding acquisition I.A.

## Competing interests

The authors declare no competing interests.
