## [Peer Review File · Nature Communications]

Stabilization of V1 interneuron-motor neuron connectivity ameliorates motor phenotype in a mouse model of ALSEditorial Note: Parts of this Peer Review File have been redacted as indicated to maintain the confidentiality of unpublished data.

REVIEWER COMMENTS

Reviewer #1 (Remarks to the Author):

In this manuscript Mora and colleagues use viral overexpression of a synaptic organiser protein to prevent a specific loss of V1 interneuron – motor neuron synapses and thus delay or prevent the motor phenotype associated with SOD1 ALS. This builds upon their recent Nature communications paper which identified the early vulnerability in this synapse type in SOD1 ALS.

Synapse loss is thought to be a conserved early feature of ALS and so the authors have an interesting rationale for performing this work. While potentially interesting for the field, there are several questions surrounding the data, methodology and text that make this feel like a preliminary report.

I will highlight my comments in order from beginning to end of the manuscript.

In Figure 1a-c the authors provide beautiful RNAScope images of endogenous Esyt1 transcripts using RNAScope. In Figure 1d, I don't believe the quantification is performed correctly. The WT analysis should go alongside the SOD1 at each time point and a 2-way ANOVA performed to test for genotype and age effects.

Figure 1g – co-localisation is near impossible to see at that level, I would recommend a high magnification insert.

Figure 1h-m – based on the models used by the authors, hEsy1 should only be expressed in En1+ve cells. However, in panel h it is clear that hEsy1 is expressed in ChAt+ve cells. How do the authors explain that, and how might it affect their interpretation of the data?

Only around 20% of En1+ve cells express the hEsy1 construct. Is it surprising that increasing Esyt1 expression in only 20% of En1+ve cells leads to significant delays in motor phenotypes? What about the other 80%? Is there a way to show that only the En1+ve/hEsy1+ve cells retain their synaptic contacts? While it is important to see the generation of hEsy1 transcripts, the manuscript lacks any evidence of protein change. The Esyt1 protein is a specific presynaptic regulator of synaptic structure so to show convincingly that increased expression of this protein leads to synaptic preservation in the SOD1 ALS model, the authors need to show robust evidence for protein expression, and preferably at the synapse.

In Figure 2a-d the authors stain for inhibitory presynapses on the surface of ChAt+ve MNs. The images are not of great quality and it's impossible to see the ChAt boundary, which makes it difficult to judge the accuracy of synaptic analyses.

In Figure 2e the authors quantify the synapse density by MN area. They show no difference in the values between WT and En1cre. Given the role of Esyt1 in synapse growth, you may expect to see an increased synaptic density compared to WT. The authors also show in 2l+m that MN number and size appear lower in the En1cre mice. The authors need to discuss these points and why En1cre-dependent hEsy1 overexpression may cause these MN changes.

It would also be interesting to know how the MN area changed (if it did) in all of the genotypes, not just En1cre.

Supplementary Figure 1 appears to be an exact replica of experiments from Supplementary Figure 1 in their previous Nature Communications paper.

Minor Comments:

There are several instances of mislabelling of figures. On page 3 line 88 the authors refer to “Figure 2f-l”. However, there are no Figure 2J or 2K panels in the figure.

On page 3 line 105 they refer to “Supplementary Figure 2e” but this is the wrong figure/panel.

Finally, this manuscript has not been formatted according to Nature Communications guidelines, so major editorial re-formatting is required.

Reviewer #2 (Remarks to the Author):

The data from Mora and colleagues show a preliminary study demonstrating a delayed motor phenotype in the ALS SOD1G93A mouse model when synapses of a particular type of inhibitory interneuron, known as V1s, are stabilized by overexpression of the presynaptic protein Esyt1. Remarkably, virus mediated expression of this protein in V1 interneurons (using En1cre mice) seem to a) retain inhibitory synapses on motor neurons (previously shown to be lost in ALS mouse models), b) protect motor neurons from cell death and c) improve behavioral motor outcomes. This manuscript follows a recent previous manuscript from these authors showing that V1 interneuron synaptic dysfunction is central to the motor phenotype displayed by ALS mouse models.

The present results using a genetic strategy to recover the phenotype are very interesting, but they seem rather preliminary. It will be important to clarify the following before publication is considered.

1. What is the significance of better preservation of motor function given that the Kaplan Meyer curve shows no significant differences in survival and loss of weight gain is no different? Does the better coordination of movement shown in Figure 3 and the better motor neuron preservation in Figure 2 correspond with maintenance of muscle innervation and force?
2. In Figure 1d there seems to be a robust trend towards less Esyt1 expression at P45. Significance was prevented because of high variability and low n (just 3 animals). Is this comparison strongly underpowered? Should n be increased. Did the authors appropriate power all their statistical comparisons? This information needs to be included in Methods.
3. Figure 1d. This figure shows Esyt1 expression associated to spinal cord cells identified by dapi nuclei. It would have been more appropriate to include a neuronal marker and even a V1 marker to assess depletion specifically in neurons and the interneurons of interest. Is the low % expression because most

cells are non-neuronal? Considering only neurons and V1s could have made this analysis more relevant and significant. Is this analysis suggesting that Esyt1 expression is reduced in ALS or that the Esyt1 expressing neurons disappeared? I am not aware of such large reductions in interneurons at these ages in the SOD1G93A model.

4. Figure 1g. I do not see the relevance of mCherry virus transfection, because Esyt1 overexpression is already tested with a viral-hEsy1 RNAscope probe. I also do not understand the orientation of figure 1g. Is this figure trying to show different segments and rostrocaudal distribution? What was the extension of expression?

5. Analyses in Fig. 1a-d seem to represent lower lumbar sections, but virus injections are performed in upper lumbar. How much did the virus spread rostrocaudally from the point of injection? Related to that, where exactly are the analyses done in the following figure 2.

6. Figure 1n. From this analysis it is impossible to know whether V1s do recover Esyt1 because there is no data about whether they lost it and this figure only tests viral derived Esyt1. Also, the authors indicate that they expressed the Esyt1 transgene in 20% of En1 neurons, but the average in the figure is lower than that. They should report it accurately. Moreover, how many V1s retain En1 expression at these ages?

7. Figure 2 analyzes synaptic densities by measuring integrated density around the motor neurons that is then normalized to area. However, the most appropriate normalization should be to perimeter.

8. It is unclear how motor neuron counts were done. Since there are large changes in size due to progression of pathology counts should be stereologically corrected, or at least count only at mid cell body established by nucleolar size cross sections, but there is no evidence any of these precautions were taken.

9. In Figure 2l, why are MN counts in wt different than en1Cre animals?

10. Interpreting the data on synaptic preservation is challenging. The dramatic reduction in VGAT coverage suggest that more than V1 synapses are affected. V1 synapses are only a proportion of the inhibitory input to the MN cell body according to current literature. Thus, the total recovery of all VGAT synapses by stabilization of only V1 synapses is difficult to interpret.

11. Reporting average angle through steps might be misleading because the same average angle could derive from a joint with a wide-angle trajectory and a joint that does not move at all, but stays fixed an angle that is average to the one with a wide motion.

12. Is the hyperflexion noted in the ankle due to larger yield because reduced body weight support and muscle strength, or because discoordination of flexor-extensor activations?

Reviewer #3 (Remarks to the Author):

This manuscript takes a really creative and promising approach to ALS interventions, attempting to preserve the microcircuit environment of motoneurons. The authors used AAV injections to overexpress the Extended Synaptotagmin gene *Esyt1* in V1 inhibitory neurons of the ventral spinal cord, with a focus on disease progression in SOD1 Tg mice. They found that *Esyt1* overexpression preserved vGAT+ boutons onto motoneurons and motoneuron number and that it ameliorated locomotor dysfunction. Overall, this work is well done and interesting but there are several points that should be addressed.

Main point:

1. The authors suggest that *Esyt1* over-expression works by stabilizing V1-MN synapses but this is not clearly shown. (1A) Is this a matter of apparent synapse preservation due to reduced MN area (as could be possible based on Figure 2m) or are there maintained synapses? (1B) Are these bona fide synapses onto MN? The ChAT staining is not very clear in the fluorescent images of Figure 2a-d, particularly for the Wt and SOD1En1Cre images. This would also be much more compelling with a post-synaptic co-localization (eg. gephyrin) to show true anatomical synapses. (1C) It would also be very interesting to know whether these vGAT+ synapses are from En1+ cells and whether other targets of En1+ cells show similar synaptic effects, though these two questions may be beyond the scope of the current work.

Minor points:

2. It is concerning that MN area is decreased upon the *Esyt1* over-expression. Is this a general effect of viral infection (is it seen with the Cherry control virus) or is it specific to chronic *Esyt1* expression?
3. Do the MN truly “shrink” or is this a shift in the size distribution of remaining MN?
4. Why did the authors change the criteria for assessing MN for Figure 2f-l (no longer using ChAT positivity)? There are certainly other large ventral neurons that could be easily mistaken for MN (eg. ventral spinocerebellar cells, border cells) and therefore size plus location do not provide a rigorous MN definition.

Misc:

5. What do the boxes in Figure 1a denote?
6. How many cells were counted per animal in Figure 1d?
7. What ages were the animals represented in Figure 1a-c?
8. The custom in situ probe for *Esyt1* should be highlighted in Figure 1f (or a version of 1f showing the recombined construct).

REVIEWER COMMENTS

We would like to thank all the reviewers for the positive assessment of our work, for finding this study interesting, and for their insightful comments. Below is the point-by-point response to their concerns and suggested experiments. All the changes can be found highlighted in red in the manuscript. The article is now edited following the Article formatting requirements.

Reviewer #1 (Remarks to the Author):

In this manuscript Mora and colleagues use viral overexpression of a synaptic organiser protein to prevent a specific loss of V1 interneuron – motor neuron synapses and thus delay or prevent the motor phenotype associated with SOD1 ALS. This builds upon their recent Nature communications paper which identified the early vulnerability in this synapse type in SOD1 ALS. Synapse loss is thought to be a conserved early feature of ALS and so the authors have an interesting rationale for performing this work. While potentially interesting for the field, there are several questions surrounding the data, methodology and text that make this feel like a preliminary report.

I will highlight my comments in order from beginning to end of the manuscript.

In Figure 1a-c the authors provide beautiful RNAScope images of endogenous Esyt1 transcripts using RNAScope. In Figure 1d, I don't believe the quantification is performed correctly. The WT analysis should go alongside the SOD1 at each time point and a 2-way ANOVA performed to test for genotype and age effects.

The WT analysis is now showed alongside the SOD1 at each timepoint as suggested, please refer to Figure 1d.

Figure 1g – co-localisation is near impossible to see at that level, I would recommend a high magnification insert.

The high magnification insert was added, and it is now shown in Figure 2c-d.

Figure 1h-m – based on the models used by the authors, hEsy1 should only be expressed in En1+ve cells. However, in panel h it is clear that hEsy1 is expressed in ChAt+ve cells. How do the authors explain that, and how might it affect their interpretation of the data?

This is an interesting point, we know from the quantifications performed in the Allodi I et al 2021 *Nature Communications* paper that there is a very small subpopulation (3-4 cells per hemi cord) of double positive cells, which express both En1 and Chat. These cells, which are obviously not motor neurons, are in the intermediate area of the spinal cord and present small soma size. We believe that they might belong to one of the Chat+ interneuron subpopulations identified by Le Pichon lab using single nucleus transcriptomics and reported in Alkaslasi MR et al 2021 *Nature Communications*. Future studies will investigate this double positive population further; however, we believe that this is outside the scope of the present manuscript, hence it will not be discussed because we do not know the physiological role of these cells and their relevance in disease. The images in Figure 2e have been adjusted to show only En1+ cells positive for Esyt1.

Only around 20% of En1+ve cells express the hEsy1 construct. Is it surprising that increasing Esyt1 expression in only 20% of En1+ve cells leads to significant delays in motor phenotypes? What about the other 80%?

The number of En1/hEsy1 positive neurons was initially quantified using the Qupath software and the DAPI staining as recommended by RNAScope, however after careful analysis of the images and the segmentation performed by the software, we realised that Qupath overestimated the number of En1+ neurons per section. This led to an underestimation of the En1/hEsy1 positive neurons. Images

were re-quantified manually, while using Neuro Trace (Fluor Nissl) as counterstaining (which stains the soma of the neurons) and the correct number of En1/hEsyt1 positive neurons within the lumbar level of the spinal cord is in average ~76%. Quantifications are shown in Figure 2k, dot plot represents the variability among the injected mice.

Is there a way to show that only the En1+ve/hEsyt1+ve cells retain their synaptic contacts?

We have performed a dual virus injection using a combination of AAV8-hSyn-DIO-hEsyt1 and AAV-phSyn1(S)-FLEX-tdTomato-T2A-SypEGFP-WPRE viruses in the En1^{cre} and SOD1^{G93A}; En1^{cre}. The AAV-phSyn1(S)-FLEX-tdTomato-T2A-SypEGFP-WPRE virus allows for visualization of the En1+/hEsyt1+ neurons and synapses since they express eGFP under the Synaptophysin promoter. Moreover, proximity ligation assay (PLA) was performed to assure colocalization of GFP and Esyt1 at synaptic level. This is shown in Figure 4. While this experiment is dependent on the efficiency of viral delivery and transfection, we can clearly show that we do have colocalization of Esyt1 and GFP. We are now also reporting VGAT and Gephyrin co-localization in Figure 5, demonstrating presence of *bona fide* inhibitory synapses upon treatment.

While it is important to see the generation of hEsyt1 transcripts, the manuscript lacks any evidence of protein change. The Esyt1 protein is a specific presynaptic regulator of synaptic structure so to show convincingly that increased expression of this protein leads to synaptic preservation in the SOD1 ALS model, the authors need to show robust evidence for protein expression, and preferably at the synapse.

Protein quantification is now included in Figure 2l-x. An antibody recognising Esyt1 and previously validated in Allodi I et al 2019 *Stem Cell Reports* was used to perform immunohistochemistry and intensity quantifications in single interneurons. Interneurons were identified by for their ventral localization and their soma size. Quantification in all four genotypes is shown in Figure 2x. Moreover, Esyt1 expression at synaptic level is shown by proximity ligation assay (PLA) performed with antibodies recognizing Esyt1 and GFP. Data is reported in Figure 4.

In Figure 2a-d the authors stain for inhibitory presynapses on the surface of ChAt+ve MNs. The images are not of great quality and it's impossible to see the ChAt boundary, which makes it difficult to judge the accuracy of synaptic analyses.

Images were retaken and are now shown in Figure 3a-d as merge and single channels. Green and red were swapped with orange and magenta. Chat is known to be downregulated in the SOD1 mice at this timepoint, so the staining for that condition is less intense.

In Figure 2e the authors quantify the synapse density by MN area. They show no difference in the values between WT and En1cre. Given the role of Esyt1 in synapse growth, you may expect to see an increased synaptic density compared to WT. The authors also show in 2l+m that MN number and size appear lower in the En1cre mice. The authors need to discuss these points and why En1cre-dependent hEsyt1 overexpression may cause these MN changes.

It would also be interesting to know how the MN area changed (if it did) in all of the genotypes, not just En1cre.

In the new version of the manuscript, we are addressing this point in detail in the Discussion paragraph (lines 404-417). We agree with the reviewer that further investigations are required to better understand Esyt1 overexpression in En1^{cre} mice and we have reached out to experts in neurosecretion both at University of Copenhagen and University of Göttingen to ask for help in dissecting the molecular mechanisms behind the observed changes.

While it is shared opinion that detrimental changes might be expected when overexpressing Esyt1 in healthy neurons, the reasons for this might be several and difficult to dissect. Esyt1 is not only important for synapses growth and maintenance, but thanks to its C2 domain is also involved in membrane tethering. Esyt1 has been shown to control the binding of ER and plasma membranes as

well as ER and mitochondria interactions, thus playing a more fundamental role also in cellular homeostasis. All these events are Ca^{2+} -dependent and changes in Esyt1 expression can lead to changes in Ca^{2+} signalling. The experiments that have been suggested would be quite difficult to perform in the mouse, and might require other experimental models, as well as a revision time that would go well beyond the reasonable extension. Moreover, another class of protein, called VAMP, and exerting similar biological functions has been previously linked in motor neuron disease (this is also reported in the discussion). For these reasons, we are not investigating further the mechanisms behind the overexpression of Esyt1 in healthy neurons in the present manuscript, but this will be addressed in future endeavours.

In addition, we know from literature and from our previous study Allodi et al 2021 Nature Communications, that motor neurons shrinkage is observed in the $\text{SOD1}^{\text{G93A}}$ mice from P84 (which is also observed for the $\text{SOD1}^{\text{G93A}}$ here), hence we believe that it might add confusion to include motor neuron areas for all genotypes.

Supplementary Figure 1 appears to be an exact replica of experiments from Supplementary Figure 1 in their previous Nature Communications paper.

Supplementary Figure 1 is not a replica, panels a and b report survival and weights of breeders between 2020-2023 (when the experiments were performed). Here, we provide an overview of the disease progression of the $\text{SOD1}^{\text{G93A}}$ mice and the double transgenic strain $\text{SOD1}^{\text{G93A}}; \text{En1}^{\text{cre}}$ (panels a and b), as well as the copy number of the mice carrying 25 copies of the $\text{SOD1}^{\text{G93A}}$ mutated gene included in the study belonging to either $\text{SOD1}^{\text{G93A}}; \text{En1}^{\text{cre}}$ or $\text{SOD1}^{\text{G93A}}$ colonies (panel c). This is included because we believe that it is fundamental to show how the colonies are maintained and their genotype and phenotype assessed to increase the reproducibility of the data. It is well known that multiple crossing might lead to changes in disease progression in the $\text{SOD1}^{\text{G93A}}$ mice and we want to highlight that our colonies are completely comparable. Several laboratories report shorter survival for the same $\text{SOD1}^{\text{G93A}}$ mice (even down to P120), other report differences between males and females that we do not observe (neither does the provider JAX), several do not assess copy number expression despite being highly recommended by the supplier. We believe that this adds confusion within the field, while decreasing data reproducibility, and we are hoping to promote further transparency on the topic.

Minor Comments:

There are several instances of mislabelling of figures. On page 3 line 88 the authors refer to “Figure 2f-l”. However, there are no Figure 2J or 2K panels in the figure.

On page 3 line 105 they refer to “Supplementary Figure 2e” but this is the wrong figure/panel.

Finally, this manuscript has not been formatted according to Nature Communications guidelines, so major editorial re-formatting is required.

These comments were addressed and mislabelling corrected. The formatting is now in line with the Article format for Nature Communications.

Reviewer #2 (Remarks to the Author):

The data from Mora and colleagues show a preliminary study demonstrating a delayed motor phenotype in the ALS $\text{SOD1}^{\text{G93A}}$ mouse model when synapses of a particular type of inhibitory interneuron, known as V1s, are stabilized by overexpression of the presynaptic protein Esyt1. Remarkably, virus mediated expression of this protein in V1 interneurons (using En1^{cre} mice) seem to a) retain inhibitory synapses on motor neurons (previously shown to be lost in ALS mouse models), b) protect motor neurons from cell death and c) improve behavioral motor outcomes. This manuscript follows a recent previous manuscript from these authors showing that V1 interneuron synaptic dysfunction is central to the motor phenotype displayed by ALS mouse models.

The present results using a genetic strategy to recover the phenotype are very interesting, but they seem rather preliminary. It will be important to clarify the following before publication is considered.

1. What is the significance of better preservation of motor function given that the Kaplan Meyer curve shows no significant differences in survival and loss of weight gain is no different?

The data reported in Supplementary Figure 1 including Kaplan Meyer and weight loss shows that the double transgenic mouse line is completely comparable with the $SOD1^{G93A}$. This is done to assure that there are no differences in progression after multiple crossing. See detailed answer to Reviewer #1 above. We apologize if this was confusing, this is now clarified in the text in lines 113-116.

Survival was not assessed after intraspinal injections, because we did not expect such localized treatment to have an impact on overall survival. However, we have now assessed survival after systemic administration of Esyt1 using an AAV-PHP.eB virus. The new data is reported in Figure 7-9. Despite the dramatic improvement of motor phenotypes shown in Figure 8 and 9, as well as the increased number of spared motor neurons in the lumbar segment of the spinal cord at end stage (Figure 9), mice did not show increased lifespan due to loss of weight (according to our ethical permit, when reaching 15% weight loss mice need to be euthanized). This could be due to the fact that, with our treatment, we might not be able to target other important circuits affected in the $SOD1^{G93A}$ model, including networks required for mastication and metabolic maintenance. This point is now clearly discussed in the Result paragraph "*Despite improved motor functions and spinal motor neuron survival, $SOD1^{G93A};En1^{cre}$ do not show increased life span due to loss of weight*" and in the Discussion at lines 364-390.

Does the better coordination of movement shown in Figure 3 and the better motor neuron preservation in Figure 2 correspond with maintenance of muscle innervation and force?

A novel algorithm called "weight support" was developed to investigate force during locomotion and implemented in our kinematics tool. Here, the ability of the mice to sustain their own weight while walking and counteract gravity was quantified. The distance of the tail from the belt of the treadmill was assessed. Here, we can see that mice overexpressing Esyt1 can support better their body, which results in their tail to be more distant from the belt. These results suggest maintenance of force after treatment. Quantifications are shown in Figure 6q and Figure 8m.

2. In Figure 1d there seems to be a robust trend towards less Esyt1 expression at P45. Significance was prevented because high variability and low n (just 3 animals). Is this comparison strongly underpowered? Should n be increased. Did the authors appropriate power all their statistical comparisons? This information needs to be included in Methods.

The number of mice included in the Esyt1 quantification was increased to 6 per condition as shown in Figure 1d. This resulted in a significant difference in expression also at P45. Information regarding power analysis is included in the Material and Methods.

3. Figure 1d. This figure shows Esyt1 expression associated to spinal cord cells identified by dapi nuclei. It would have been more appropriate to include a neuronal marker and even a V1 marker to assess depletion specifically in neurons and the interneurons of interest. Is the low % expression because most cells are non-neuronal? Considering only neurons and V1s could have made this analysis more relevant and significant. Is this analysis suggesting that Esyt1 expression is reduced in ALS or that the Esyt1 expressing neurons disappeared? I am not aware of such large reductions in interneurons at these ages in the $SOD1^{G93A}$ model.

New quantifications of $En1$ /Esy1 double positive neurons were performed at P45, P63 and P84 timepoints and are now reported in Figure 1e-g. Esyt1 is downregulated in V1 interneurons from P63. We observe Esyt1 expression mainly in neurons. Esyt1 appears downregulated in multiple neuronal types within the ventral horn of the spinal cord including motor neurons, so it is not a V1 specific

loss.

4. Figure 1g. I do not see the relevance of mCherry virus transfection, because Esyt1 overexpression is already tested with a viral-hEyt1 RNAscope probe. I also do not understand the orientation of figure 1g. Is this figure trying to show different segments and rostrocaudal distribution? What was the extension of expression?

The mCherry virus was included to investigate the spread of the transfection. Figure 2c shows a tile scan of a longitudinal section of the lumbar spinal cord. DH and VH indicate dorsal horn and ventral horn respectively. The extension of expression was approximately 1.2 mm rostral and caudal spread from injection site. This is now clearly stated in the manuscript at line 123-124. A panel showing higher magnification has also been included in Figure 2c.

5. Analyses in Fig. 1a-d seem to represent lower lumbar sections, but virus injections are performed in upper lumbar. How much did the virus spread rostrocaudally from the point of injection? Related to that, where exactly are the analyses done in the following figure 2.

Lumbar section in Figure 1a shows a lumbar segment 4 section. Quantifications were performed in lumbar segments L1-L4 also for synaptic connectivity and motor neuron quantifications. Serial sections were collected in series of 10 with 5/6 slices of spinal cord (12 μ m thickness) per slide. Three slides were stained per mouse; hence quantifications were performed on a wider portion of the lumbar segment of the spinal cord, and not only the upper lumbar region. This information is now clarified in the Material and Methods section at line 567-569.

6. Figure 1n. From this analysis it is impossible to know whether V1s do recover Esyt1 because there is no data about whether they lost it and this figure only tests viral derived Esyt1. Also, the authors indicate that they expressed the Esyt1 transgene in 20% of En1 neurons, but the average in the figure is lower than that. They should report it accurately. Moreover, how many V1s retain En1 expression at these ages?

As mentioned above, Esyt1 was quantified in V1 interneurons and is downregulated from P63. Quantification of Esyt1 protein was also performed and showed recovery of Esyt1 levels in interneurons after viral delivery. Quantifications are now reported in Figure 2l-x. As also mentioned above in our reply to Reviewer #1 the number of En1/hEyt1 positive neurons was previously quantified using Qupath software and the DAPI staining as recommended by RNAscope, but this led to an overestimation of En1+ cells and an underestimation of the En1/hEyt1 positive neurons. Upon manual quantification and Fluor Nissl counterstaining the number of En1/hEyt1+ is 76% on average. Variability among mice can be observed in the graph shown in Figure 2k.

7. Figure 2 analyzes synaptic densities by measuring integrated density around the motor neurons that is then normalized to area. However, the most appropriate normalization should be to perimeter.

The normalization is now performed by perimeter and reported in Figure 3e and modified accordingly in the text (Results and Material and Methods).

8. It is unclear how motor neuron counts were done. Since there are large changes in size due to progression of pathology counts should be stereologically corrected, or at least count only at mid cell body established by nucleolar size cross sections, but there is no evidence any of these precautions were taken.

Only motor neurons showing mid cell body and clear nucleus were included in the study. This is now clearly stated in the Results and it is reported in Material and Methods at line 585-586.

9. In Figure 2l, why are MN counts in wt different than en1Cre animals?

Only neurons within lamina IX, with clear motor neuron shape and nucleus, and a diameter of >28 μm were included in the quantification. This is because we aimed to focus mainly on the large size motor neurons (putative fast motor neurons). In our previous work, Allodi I et al 2021 *Nature Communications*, we showed that V1 interneurons had stronger inputs on the fast motor neurons, hence we wanted to assess if our treatment could retain fast motor neurons. Fast motor neurons can be identified by the MMP9 marker, which was not used here because the marker is lost at later stages of disease. However, when we quantified motor neurons in En1cre mice overexpressing Esyt1 we found a lower number of >28 μm diameter motor neurons. This led us to look at potential motor neuron shrinkage upon Esyt1 overexpression in the healthy mice. We found that the size of the motor neuron shrunk in this condition. We discuss this point in detail in the response to Reviewer #1, the reasons might be multiple due to the pivotal role played by Esyt1 in cellular homeostasis, and not only at synaptic level. Further investigations will be performed in the future to understand the mechanisms behind these results.

10. Interpreting the data on synaptic preservation is challenging. The dramatic reduction in VGAT coverage suggest that more than V1 synapses are affected. V1 synapses are only a proportion of the inhibitory input to the MN cell body according to current literature. Thus, the total recovery of all VGAT synapses by stabilization of only V1 synapses is difficult to interpret.

The synaptic density quantifications in Figure 3 are performed at P112, a timepoint in which SOD1 mice would normally show overt symptoms and a significant amount of motor neuron loss. At this timepoint, the dramatic reduction of VGAT observed at P112 is most likely not only due to V1 interneurons (e.g., V2b interneurons which are also inhibitory). We now know from our new data Montañana-Rosell et al <https://doi.org/10.1101/2023.09.17.558103> that more neural populations are affected at P112 timepoint. We agree with the reviewer that the effect of V1 stabilization is substantial, and we believe that other cell types within the network might benefit from V1 stabilization as well – not only the motor neurons. From the dual injections reported in Figure 4, we can observe GFP and PLA staining also on other cells found in the ventral and intermediate areas of the spinal cord (Figure 4b). Thus, VGAT recovery would not only come from V1 interneurons. We are currently investigating the effects of V1 degeneration and stabilization at network level using computational modelling. This work is based on a model initially developed by the Rybak lab (Shevtsova et al 2022 *J Mol Sci*), where we focus on a unilateral central pattern generator coordinating flexor-extensor alternation. This data is shown as reviewer only figure since it is still preliminary. Here, we observe that loss of V1 interneurons destabilizes not only motor neurons, but also other interneuron populations including excitatory neurons (e.g., V2a). This event can be rescued by stabilization of V1 synapses during early symptomatic stages (P63), when few neurons are lost. Since our treatment is delivered before loss of connectivity, we believe that we can retain a larger number of V1 synapses, not only on motor neurons but also on other neurons, exerting a more extensive effect on the network. My lab is currently investigating this with a combination of computational modelling and transcriptomic analysis.

[REDACTED]

11. Reporting average angle through steps might be misleading because the same average angle could derive from a joint with a wide-angle trajectory and a joint that does not move at all, but stays fixed an angle that is average to the one with a wide motion.

Here, the difference between the maximum and minimum angles within each step cycle was estimated, and the average joint angles per mouse were reported. This is done to avoid reporting misleading values. We are sorry that was not clear. It is not stated in line 241.

12. Is the hyperflexion noted in the ankle due to larger yield because reduced body weight support and muscle strength, or because discoordination of flexor-extensor activations?

V1 and V2b inhibitory interneurons were described to contribute to flexor-extensor coordination by the Goulding lab (Zhang et al 2014 *Neuron*). In another publication, the same lab (Britz et al 2015 *eLife*) found that after silencing of V1 interneurons mice show hyperflexion of the hindlimbs. In our 2021 paper (Allodi I et al *Nature Communications*), we show that the same hyperflexion is found in the SOD1^{G93A} mice. So, this phenotype is due to the discoordination of flexor-extensor activations. This is now stated more clearly in the manuscript at lines 360-361, including relevant citations.

Reviewer #3 (Remarks to the Author):

This manuscript takes a really creative and promising approach to ALS interventions, attempting to preserve the microcircuit environment of motoneurons. The authors used AAV injections to overexpress the Extended Synaptotagmin gene *Esy1* in V1 inhibitory neurons of the ventral spinal cord, with a focus on disease progression in SOD1 Tg mice. They found that *Esy1* overexpression preserved vGAT+ boutons onto motoneurons and motoneuron number and that it ameliorated locomotor dysfunction. Overall, this work is well done and interesting but there are several points that should be addressed.

Main point:

1. The authors suggest that *Esy1* over-expression works by stabilizing V1-MN synapses but this is not clearly shown. (1A) Is this a matter of apparent synapse preservation due to reduced MN area (as could be possible based on Figure 2m) or are there maintained synapses?

In our previous study Allodi et al 2021 *Nature Communications*, we show that 50% of glycinergic inputs were lost at P45 timepoint in the SOD1^{G93A} mice. We also reported loss of *En1* transcript from P63, suggesting V1 interneuron degeneration. In the present study, *Esy1* is overexpressed in V1 interneurons early in disease (P30) with the aim of preserving synaptic inputs. Here, we show that *Esy1* is downregulated in *En1* positive neurons from P63, but that upon AAV8-hSyn-DIO-h*Esy1* transfection, *Esy1* protein level is maintained in interneurons. Altogether this data suggests that

synapses are maintained. We exclude that the effect observed is due to reduced motor neuron perimeter, as shown in the figure below comparing motor neuron perimeter in WT and SOD1^{G93A};En1^{cre} mice. Here, we use the perimeter instead of the area as suggested by Reviewer #2.

(1B) Are these bona fide synapses onto MN? The ChAT staining is not very clear in the fluorescent images of Figure 2a-d, particularly for the Wt and SOD1En1Cre images. This would also be much more compelling with a post-synaptic co-localization (eg. gephyrin) to show true anatomical synapses.

We thank the reviewer for suggesting this experiment, we now know that these are *bona fide* synapses. We have quantified VGAT and Gephyrin co-localization on motor neurons in all conditions after Esyt1 overexpression, quantifications are shown in Figure 5. We have also changed the ChAT-VGAT images in now Figure 3a-d, to make the staining clearer.

(1C) It would also be very interesting to know whether these vGAT+ synapses are from En1+ cells and whether other targets of En1+ cells show similar synaptic effects, though these two questions may be beyond the scope of the current work.

To address this concern, we have injected a mix of AAV8-hSyn-DIO-hEsy1 and AAV-phSyn1(S)-FLEX-tdTomato-T2A-SypEGFP-WPRE viruses in the En1^{cre} and SOD1^{G93A}; En1^{cre}, and then performed PLA for GFP and Esyt1 proteins to validate co-localization at synaptic level. The AAV-phSyn1(S)-FLEX-tdTomato-T2A-SypEGFP-WPRE virus visualizes synaptic terminals of transduced neurons in a cre-dependent manner. This experiment validates the expression of Esyt1 at synaptic level on motor neurons in the En1^{cre} and SOD1^{G93A} mice. As shown in Figure 4b, we observe GFP positive synapses on other cells that are not motor neurons within the ventral and intermediate areas of the spinal cord, as well as on processes. This suggests that Esyt1 overexpression might be acting also on other V1 targets.

Minor points:

2. It is concerning that MN area is decreased upon the Esyt1 over-expression. Is this a general effect of viral infection (is it seen with the Cherry control virus) or is it specific to chronic Esyt1 expression? This seems to be specific for Esyt1 overexpression in healthy interneurons. Please, see detailed information added in Results and Discussion paragraphs.

3. Do the MN truly “shrink” or is this a shift in the size distribution of remaining MN?

We do believe that the motor neurons are shrinking because we did not observe changes in motor phenotype in these mice. In case of shift in motor neuron size distribution, we would have a majority of intermediate and slow motor neurons left within the lumbar segments of the spinal cord, and we would expect changes in gait, kinematics, or weight support that we did not observe (at least at these timepoints). For more detailed response, please refer to Reviewer #1.

4. Why did the authors change the criteria for assessing MN for Figure 2f-l (no longer using ChAT positivity)? There are certainly other large ventral neurons that could be easily mistaken for MN (eg. ventral spinocerebellar cells, border cells) and therefore size plus location do not provide a rigorous

MN definition.

ChAT is downregulated in SOD1^{G93A} motor neurons at the timepoints included in the study (from P84) due to neuromuscular junction denervation. This is widely reported in the ALS literature. So, to avoid underestimating the number of motor neurons in the SOD1^{G93A} mouse we used Nissl staining instead of ChAT. This is a standard procedure that we used in previous studies Allodi et al 2021 *Nature Communications*, Nizzardo M et al 2020 *Acta Neuropathologica*, Allodi I et al 2016 *Scientific Reports* and that is used by several laboratories.

Misc:

5. What do the boxes in Figure 1a denote?

The boxes in Figure 1a show the ventral area selected for Esyt1 quantification. This is now clearly explained in Figure 1 legend.

6. How many cells were counted per animal in Figure 1d?

Between 600-700 neurons were counted per condition per timepoint, it is now reported in the text at line 94.

7. What ages were the animals represented in Figure 1a-c?

Figure 1a-c show Esyt1 expression in P84 mice.

8. The custom in situ probe for Esyt1 should be highlighted in Figure 1f (or a version of 1f showing the recombined construct).

Custom probe targeting of recombined construct was added in Figure 2l.

REVIEWER COMMENTS

Reviewer #1 (Remarks to the Author):

Mora and colleagues should be commended for taking the time to significantly expand the work and address reviewers' comments. This is a much larger and detailed study than the first version and the introduction of a systemic viral approach is particularly interesting. My points about synaptic level protein change have been addressed nicely with new PLA experiments and suggestions for improved data visualisation have strengthened the manuscript, in my opinion.

My initial review has been largely successfully rebutted, with only a few minor points to mention.

1 – I believe the data in Figure 1d and 1g should be analysed by 2-way ANOVA as both time point and genotype are variables. See your new analysis in Figure 6 as an example.

2 – Figures 2e-k show a robust increase in hESYT1 transcript in En1cre mice, with no expression in WT. However, this doesn't carry over into your protein data in Figure 2y as no difference is seen between WT and En1cre. Can you explain the discrepancy?

3 – In Figure 2y and Figure 5u+v, I feel by plotting data from individual neurons you are pseudo-replicating. I realise n-numbers are low, but the correct way to analyse would be to pool the individual cells from each animal to generate a mean intensity for each mouse.

4 – I really like Figure 7 and the data within, I would just like to see some higher mag examples of c,d,e and h either within this Figure or as supplementary.

5 – You nicely discuss the paradox between preserved motor function and lack of delay to death following systemic treatment. Given the preservation of motor function until at least P112 it suggests the weight loss is not due to inability to access food and you suggest it may be due to reduced mastication etc. Is it possible to measure food intake to see if it is a physical problem with food intake, rather than an alteration in metabolism? It would be interesting to know if your systemic treatment also delays/prevents this physical problem and would provide important information on what treatments to combine in the future.

Reviewer #2 (Remarks to the Author):

This is a very interesting manuscript and of high importance. It demonstrates the importance of a major inhibitory input onto motoneurons during progression of ALS using a rescuing approach. The authors performed some clever experiments using interesting viral based approaches (that might be of

translational value) to reduce the loss of these synapses. The authors show how this manipulation decreases motor function dysfunction in the Sod1G93A mouse model of ALS, although unfortunately survival rates were not affected.

The response to the previous review was extraordinary and the manuscript is much improved. It also increased in scope and significance by the addition of many new experiments. Unfortunately, these new experiments raised new issues that need to be clarified. But overall, the manuscript is very much improved, and my recommendations are all related to presentation and interpretation.

Overall comment that needs discussion:

The reduction of VGAT synapses on the cell bodies of motoneurons is ascribed mainly to V1 inputs, however published literature suggests that V1 interneurons provided a little over half of the synapses in lumbar LMC motoneurons (Zhang et al., 2014). Are non-V1 VGAT synapses also affected and rescued by Eysyt1 overexpression in V1 interneurons?

1. Figure 2 legend. Quantification in panel y: The description of the data indicates “N=2-4 mice per conditions: 10-20 motor neurons analyzed per condition”. Do you mean interneurons? Also, these n’s do not correspond with the number of dots in the graph per condition. Please clarify.

2. Criteria for “fast” motoneuron identification based on a diameter larger than 28 microns might not be accurate. Differences in motoneuron cell body size are not as clear cut in mice as they are in cats where they were first described. If the authors can support this criterion with a reference that will be OK, but if they can’t it might be better to obviate this distinction such that that non-well supported criteria are not spread in the field. I predict this study will have a significant impact in the future and it will be a pity if it includes statements that a the less informed reader might interpret as well-established. Nowhere in the manuscript it becomes relevant what type of alpha motoneuron is studied.

On the other hand, these criteria can be used to differentiate gamma vs alpha motoneurons in mice. This is very appropriate given the known reduced density of inhibitory synapses on gamma motoneurons and that if included will confuse and increase variability in the results.

3. Figure 4. The construct in Figure 4a does not match the construct described in the results. Is it a Flex or DIO construct?

4. Figure 5. This figure shows no gephyrin immunoreactivity and almost no VGAT in SOD1 mice, either on the motoneuron or anywhere in the neuropil. This is very surprising and does not represent the quantitative data in Figures 5u. The data show a depletion of a bit over 50% in the area with VGAT and gephyrin co-localization, not an almost total depletion. Please select images for illustration that represent better quantitative data.

5. Figure 6K-n. Please explain in the figure and results at which point in the step cycle are the angles measured. Is this maximum flexion or referred to a particular point in the cycle? Then in the companion supplementary figure 3 the data shows more a lack of extension than hyperflexion. Please explain the conclusion of an hyperflexed phenotype based on the data presented.

6. Figure 7. Although no effects were detected in the behavioral tests implemented after retroorbital expression of AAV-PHP.eb vector I was surprised to see expression in cortex, where no engrailed-1 expression occurs, to my knowledge. Therefore off-target expression cannot be completely ruled out, although it is clear that V1 interneurons in the spinal cord are preferentially targeted. The authors should be more conservative in the conclusion of possible off-target effects.

7. Throughout the manuscript, please remember that data is plural. These data instead of “this data”.

Reviewer #3 (Remarks to the Author):

The authors have addressed my concerns

REVIEWER COMMENTS

Reviewer #1 (Remarks to the Author):

We would like to thank the reviewer for the positive feedback on our resubmission and for the further comments raised. Below is our point-by-point response.

Mora and colleagues should be commended for taking the time to significantly expand the work and address reviewers' comments. This is a much larger and detailed study than the first version and the introduction of a systemic viral approach is particularly interesting. My points about synaptic level protein change have been addressed nicely with new PLA experiments and suggestions for improved data visualisation have strengthened the manuscript, in my opinion.

My initial review has been largely successfully rebutted, with only a few minor points to mention.

1 – I believe the data in Figure 1d and 1g should be analysed by 2-way ANOVA as both time point and genotype are variables. See your new analysis in Figure 6 as an example. The data in Figure 1d and 1g is analysed by 2-way ANOVA, it was miswritten, and it has been changed.

2 – Figures 2e-k show a robust increase in hESYT1 transcript in En1cre mice, with no expression in WT. However, this doesn't carry over into your protein data in Figure 2y as no difference is seen between WT and En1cre. Can you explain the discrepancy? The difference between Figure 2e-k and 2y, is that the RNAscope probe recognises only the specific expression of the virus, while the antibody recognises all Esyt1 protein present in the neurons, including the endogenous Esyt1 expressed in control conditions in the WT and En1cre mice. This is the reason why there is also expression in the WT. This is now clarified in the text – line 134-135.

3 – In Figure 2y and Figure 5u+v, I feel by plotting data from individual neurons you are pseudo-replicating. I realise n-numbers are low, but the correct way to analyse would be to pool the individual cells from each animal to generate a mean intensity for each mouse.

The quantification of individual neurons was pooled, and it is now showed by mouse. The post hoc analysis was changed to Fisher's LSD due to the low n-numbers.

4 – I really like Figure 7 and the data within, I would just like to see some higher mag examples of c,d,e and h either within this Figure or as supplementary.

Higher magnification images were added to Figure 7 for spinal cord, midbrain and cortex, and are now shown in panels d, f and h. The RNAscope image showing higher magnification and co-localization of probes is now shown in Figure 7k.

5 – You nicely discuss the paradox between preserved motor function and lack of delay

to death following systemic treatment. Given the preservation of motor function until at least P112 it suggests the weight loss is not due to inability to access food and you suggest it may be due to reduced mastication etc. Is it possible to measure food intake to see if it is a physical problem with food intake, rather than an alteration in metabolism? It would be interesting to know if your systemic treatment also delays/prevents this physical problem and would provide important information on what treatments to combine in the future.

The reviewer is right, and we have thought about this. We initially planned to include a food intake paradigm after systemic administration. However, we ran into the issue that our mice are housed by litters and not by genotype. Hence, there are both healthy and mutant mice in the same cage, which makes it impossible to efficiently measure food intake. We thought of housing the mice individually for a short amount of time and look at food intake, but from the small pilot we ran with WT mice, and we saw that this assessment increased stress levels. Thus, we could not apply this paradigm to our treated SOD1 mice, because this would have increased the overall severity of the experiments, and that was not covered by our current ethical permit. Our plan is to perform these experiments in the future under a revised ethical permit.

Reviewer #2 (Remarks to the Author):

We would like to thank the reviewer for these very positive comments. Below is our response to the further concerns.

This is a very interesting manuscript and of high importance. It demonstrates the importance of a major inhibitory input onto motoneurons during progression of ALS using a rescuing approach. The authors performed some clever experiments using interesting viral based approaches (that might be of translational value) to reduce the loss of these synapses. The authors show how this manipulation decreases motor function dysfunction in the Sod1G93A mouse model of ALS, although unfortunately survival rates were not affected.

The response to the previous review was extraordinary and the manuscript is much improved. It also increased in scope and significance by the addition of many new experiments. Unfortunately, these new experiments raised new issues that need to be clarified. But overall, the manuscript is very much improved, and my recommendations are all related to presentation and interpretation.

Overall comment that needs discussion:

The reduction of VGAT synapses on the cell bodies of motoneurons is ascribed mainly to V1 inputs, however published literature suggests that V1 interneurons provided a little over half of the synapses in lumbar LMC motoneurons (Zhang et al., 2014). Are non-V1 VGAT synapses also affected and rescued by Esyt1 overexpression in V1 interneurons? In Allodi et al 2021 *Nature Communications*, we showed that the loss of synapses on motor neurons at P63 timepoint in the SOD1G93A mice was mainly linked to the V1 degeneration. This was proven at anatomical and functional levels. However, in the

present study, we are using later timepoints (P112 and later) with the aim of investigating the changes in symptoms after *Esyt1* overexpression. Considering that in the 2021 paper we observed 50% reduction of GlyT2+ synapses at P45 and in this study, we report 80% reduction of VGAT+ synapses at P112 it is possible that, at later stages, also other populations are involved. Our new results (Montañana-Rosell et al 2023 BioRxiv) show that degeneration of spinal networks is exacerbated at P112 and V2a interneurons are also affected. However, we do not know the fate of other inhibitory populations (e.g., V2b and V0d) at later stages of disease, and we cannot exclude their contribution to the reduction of VGAT synapses observed at P112. We will investigate the fate of V2b and V0d in disease in future studies. For now, we discuss this point both in the Results and Discussion paragraphs.

After *Esyt1* overexpression, it is clear from the quantifications reported in Figure 3e and Figure 5 that the number of VGAT+ synapses is comparable to control conditions. Hence, *Esyt1* overexpression in V1 interneurons most likely has an effect on the spinal network and not only on the motor neurons (as we also stated in our previous point-by-point response). We are now discussing this further in the Discussion paragraph – line 342-352.

1. Figure 2 legend. Quantification in panel y: The description of the data indicates “N=2-4 mice per conditions: 10-20 motor neurons analyzed per condition”. Do you mean interneurons? Also, these n’s do not correspond with the number of dots in the graph per condition. Please clarify.

Yes, that is a typo. The correct sentence is “10-20 interneurons analysed per condition”. This was changed in the figure legend. Also, the quantifications are now showed per mouse as suggested by Reviewer #1.

2. Criteria for “fast” motoneuron identification based on a diameter larger the 28 microns might not be accurate. Differences in motoneuron cell body size are not as clear cut in mice as they are in cats where they were first described. If the authors can support this criterion with a reference that will be OK, but if they can’t it might be better to obviate this distinction such that that non-well supported criteria are not spread in the field. I predict this study will have a significant impact in the future and it will be a pity if it includes statements that a the less informed reader might interpret as well-established. Nowhere in te manuscript it becomes relevant what type of alpha motoneuron is studied.

On the other hand, these criteria can be used to differentiate gamma vs alpha motoneurons in mice. This is very appropriate given the known reduced density of inhibitory synapses on gamma motoneurons and that if included will confuse and increase variability in the results.

We understand the reviewer’s concern and we have now removed the statement regarding the fast motor neurons. The revised manuscript only refers to motor neurons.

3. Figure 4. The construct in Figure 4a does not match the construct described in the results. Is it a Flex or DIO construct?

The construct in Figure 4a has been modified to match the results. The *SypEGFP* is a FLEX construct.

4. Figure 5. This figure shows no gephyrin immunoreactivity and almost no VGAT in SOD1 mice, either on the motoneuron or anywhere in the neuropil. This is very surprising and does not represent the quantitative data in Figures 5u. The data show a depletion of a bit over 50% in the area with VGAT and gephyrin co-localization, not an almost total depletion. Please select images for illustration that represent better quantitative data. New images have been included to Figure 5 to better represent the quantifications.

5. Figure 6K-n. Please explain in the figure and results at which point in the step cycle are the angles measured. Is this maximum flexion or referred to a particular point in the cycle?

The angles reported in Fig6 (k-n) are not from a specific point within a step-cycle; it is the difference between the maximum and the minimum inflection angle per joint within a step cycle. This difference angle is then averaged over 15 step cycles per mouse. This is currently described in the caption for Supplementary Figure 3 and it has also been clarified in the Methods section in the "Kinematic Analysis" subsection.

Then in the companion supplementary figure 3 the data shows more a lack of extension than hyperflexion. Please explain the conclusion of an hyperflexed phenotype based on the data presented.

Panels p-t in supplementary figure 3 were changed to more representative examples of the amelioration observed after Esyt1 overexpression in the SOD1^{G93A}; En1^{cre} mice. Britz O et al 2015 eLife reported reduction in the ankle angle after ablation of V1 interneurons, this phenotype was described as hyperflexion of the hindlimb during early swing phase. This was supported by EMG recordings showing expansion of flexor muscle activity. Similarly, in Allodi et al 2021 we observed reduction of the ankle angle in the SOD1^{G93A} which recapitulated the phenotype upon V1 ablation. Thus, we refer to it as hyperflexion. In the present work, we observe an increase in the ankle angle after Esyt1 overexpression in the SOD1^{G93A}; En1^{cre} mice. However, the analysis of the hindlimb kinematics performed at P112 shows changes in all joint angles, including the hip angle, a phenotype that was not observed at early symptomatic stages in Allodi et al 2021. This might be characteristic of a more advanced stage of disease; however, we do not know its direct cause. While we observe a recovery of all the other joint angles (knee, ankle, and foot) which can be linked to V1 connectivity restoration as shown in Figure 6 and Supplementary Figure 3, the hip angle cannot be restored by Esyt1 treatment. This is now clearly stated in the Results and in the Discussion.

6. Figure 7. Although no effects were detected in the behavioral tests implemented after retroorbital expression of AAV-PHP.eb vector I was surprised to see expression in cortex, where no engrailed-1 expression occurs, to my knowledge. Therefore off-target expression cannot be completely ruled out, although it is clear that V1 interneurons in the spinal cord are preferentially targeted. The authors should be more conservative in the conclusion of possible off-target effects.

This point has been corrected in the manuscript.

7. Throughout the manuscript, please remember that data is plural. These data instead of "this data".

Thank you for pointing this out, this has been changed throughout the manuscript.

Reviewer #3 (Remarks to the Author):

We would like to thank the reviewer for the positive evaluation of our revised manuscript.

The authors have addressed my concerns

REVIEWERS' COMMENTS

Reviewer #1 (Remarks to the Author):

The authors have adequately addressed my comments.

Reviewer #2 (Remarks to the Author):

The authors have addressed all concerns